# The effect of temperature-dependent material properties on simple thermal models of subduction zones

Iris van Zelst[1,2], Cedric Thieulot[3], and Timothy J. Craig[1]

[1]Institute for Geophysics and Tectonics, School of Earth and Environment, University of Leeds, Leeds, LS2 9JT, United Kingdom
[2]Institute of Planetary Research, German Aerospace Center (DLR), Berlin, Germany
[3]Department of Earth Sciences, Utrecht University, Utrecht, The Netherlands

**Correspondence:** Iris van Zelst (iris.vanzelst@dlr.de / iris.v.zelst@gmail.com)

**Abstract.**

To a large extent, the thermal structure of a subduction zone determines where seismicity occurs through controls on the transition from brittle to ductile deformation and the depth of dehydration reactions. Thermal models of subduction zones can help understand the distribution of seismicity by accurately modelling the thermal structure of the subduction zone. Here, we assess a common simplification in thermal models of subduction zones, i.e., constant values for the thermal parameters. We use temperature-dependent parameterisations, constrained by lab data, for the thermal conductivity, heat capacity, and density, to systematically test their effect on the resulting thermal structure of the slab. To isolate this effect, we use the well-defined, thoroughly studied, and highly simplified model setup of the subduction community benchmark by van Keken et al. (2008) in a 2D finite element code. To ensure a self-consistent and realistic initial temperature-profile for the slab, we implement a 1D plate model for cooling of the oceanic lithosphere with an age of 50 Myr instead of the previously used half-space model. Our results show that using temperature-dependent thermal parameters in thermal models of subduction zones affects the thermal structure of the slab with changes on the order of tens of degrees and hence tens of kilometres. More specifically, using temperature-dependent thermal parameters results in a slightly cooler slab with e.g., the 600°C isotherm reaching almost 30 km deeper. From this, we infer that these models would predict a larger estimated seismogenic zone and a larger depth at which dehydration reactions responsible for intermediate-depth seismicity occur. We therefore recommend that thermo(-mechanical) models of subduction zones take temperature-dependent thermal parameters into account, especially when inferences on seismicity are made.

## 1 Introduction

The thermal structure of subduction zones plays a vital role in controlling many geological and petrological processes, including the dehydration of the subducting plate (Peacock, 2001; Hacker et al., 2003a), the subsequent hydration of the mantle and overriding plate (Peacock, 1993a; Abers et al., 2017), and mineralogical variations, including serpentinisation (Hyndman and Peacock, 2003). Furthermore, seismicity can often be related to both the thermal structure, and to various processes controlled by the pressure and temperature evolution of the slab (Scholz, 2019). For example, intermediate-depth earthquakes are asso-

ciated with a process called dehydration embrittlement (e.g., Green and Houston, 1995; Peacock, 2001; Hacker et al., 2003b;
Yamasaki and Seno, 2003; Jung et al., 2004; Wang et al., 2017), in which water is released during the compositional evolution
of the slab, as hydrous minerals progressively transform to less hydrous phases (e.g., from blueschist to eclogite (van Keken
et al., 2011)). The addition of free fluids to the system acts against the pressure of the surrounding rock, permitting earthquakes
to occur at depths where the confining pressure is otherwise too large. These phase transitions are linked to specific temperature
and pressure conditions, suggesting that a thorough grasp of those conditions at depth could indicate where intermediate-depth
seismicity would be likely to occur (e.g., Hacker et al., 2003a). Similarly, megathrust earthquakes occur within the seismogenic
zone, the downdip limit of which is thought to be the transition from brittle to ductile deformation (Peacock and Hyndman,
1999; Scholz, 2019), and is again controlled, directly or indirectly, by temperature, with isotherms of $350 - 450°C$ typically
linked to this change (Hyndman and Wang, 1993; Hyndman et al., 1997; Gutscher and Peacock, 2003).

From these examples, it becomes clear that it is important to have a thorough understanding of the thermal structure of a slab
in order to better understand the distribution of the full spectrum of seismicity associated with the subduction process. However,
it is hard to obtain direct observational data on the thermal structure of the slab, due to the inaccessibility of subduction zones
and the difficulty of obtaining data at great depths (i.e., larger than 10 km).

The dependence of seismic wavespeeds on temperature allows seismic tomography studies to give a broad overview of
the large-scale thermal structure of the subduction zone as a whole, but such studies typically lack the resolution to infer
the thermal structure of the slab itself in great detail (e.g., Abers et al., 2006; Pozgay et al., 2009). In addition, the observed
velocity anomalies in tomographic models are not exclusively due to temperature, and wavespeed variations are also related to
other factors, particularly composition, density, mineralogy, and the presence of fluids (e.g., Hacker et al., 2003a; Blom et al.,
2017). Whilst bore-hole experiments and marine heat flow measurements can provide vital insights into the thermal state of the
shallow seismogenic zone (e.g., Hyndman and Wang, 1993; Chang et al., 2010; Fulton et al., 2013; Harris et al., 2013; Yabe
et al., 2019), such measurements are extremely local and fail to give a good overview of the conditions of the subduction zone
as a whole, especially the finer details of the temperature structure within the slab.

In light of the limited available data on the thermal structure of subduction zones, geodynamic numerical modelling provides
a way of investigating the complete temperature field of subduction zones in relation to the thermal and dynamic evolution of
the slab (see Peacock, 2020, for an overview). The starting point for thermal models of subduction zones are one-dimensional
models of the cooling of oceanic lithosphere that define the thermal structure of the slab for a certain plate age, including half-
space cooling models and more advanced plate models (McKenzie and Sclater, 1969; Parsons and Sclater, 1977; Stein and
Stein, 1994; Hillier and Watts, 2005; McKenzie et al., 2005; Crosby et al., 2006; Emmerson and McKenzie, 2007; Richards
et al., 2018). Extending this thermal modelling to two dimensions to study the thermal evolution of a subduction zone has
provided insights into the predicted location of dehydration and melting processes linked to intermediate-depth seismicity
(Ponko and Peacock, 1995; Peacock and Wang, 1999; van Keken et al., 2002; Abers et al., 2006; Syracuse et al., 2010; van
Keken et al., 2012, 2019). Apart from pure thermal models, thermo-mechanical models with various complexities such as
melting and dehydration reactions have also been employed (e.g., Gerya and Meilick, 2011; Gerya, 2011; Faccenda et al.,
2012; Arcay, 2017; Beall et al., 2021), leading to insights into subduction dynamics and estimates of the depth of intermediate-

depth seismicity and the geometry of the megathrust. When these types of models additionally account for an inertia term in so-called seismo-thermo-mechanical models (van Dinther et al., 2013b, a), megathrust slip events are resolved allowing for estimates of the maximum size of the seismogenic zone and the distribution of seismicity in a given subduction geometry (van Dinther et al., 2014; Van Zelst et al., 2019; Petrini et al., 2020; Brizzi et al., 2020). These types of modelling have the advantage that the temperature can be calculated across the entire subduction zone with arbitrary resolution. However, their results depend on their initial and boundary conditions and the assumptions that enter the models at various stages (Van Zelst et al., 2022).

Numerical models of the temperature structure of subduction zones are subject to a range of simplifications. One, which we seek to address here, is that the thermal parameters in the model, i.e., the thermal conductivity, heat capacity, and density, are commonly assumed to be constant or merely material-dependent. In contrast, laboratory experiments have shown that these parameters actually depend on temperature and can differ as much as a factor of 2 depending on the temperature (e.g., Berman, 1988; Berman and Aranovich, 1996; Seipold, 1998; Hofmeister, 1999; Xu et al., 2004; Wen et al., 2015; Su et al., 2018). The inclusion of such parameters into models for the cooling of oceanic lithosphere has made a significant difference to both the resulting thermal structure, and its interpretation and implications (Denlinger, 1992; McKenzie et al., 2005; Richards et al., 2018). Previous one-dimensional (Emmerson and McKenzie, 2007) and three-component slab (Chemia et al., 2015) studies have highlighted the potential for a similar impact on the more complex thermal structure of subduction zones.

Given the sensitivity of the various processes mentioned above to small-scale variations in the temperature evolution of the slab, we therefore seek to quantify the potential impact of the temperature-dependence of thermal parameters on subduction zone thermal structure, and to build towards their routine incorporation.

In order to assess the effect of temperature-dependent thermal parameters on the resulting thermal structure of the slab, we perform a systematic study by using the well-defined, highly simplified setup of the subduction community benchmark by van Keken et al. (2008) with the addition of temperature-dependent functions for the thermal conductivity, heat capacity, and density as constrained by laboratory experiments (Section 2). We show that using temperature-dependent parameters in geodynamic models changes the resultant thermal structure of the slab, relative to models with fixed values (Section 3). To relate this change in thermal structure to expected seismicity and mineralogical changes in the slab, we discuss the change in the expected depth of intermediate-depth seismicity when temperature-dependent thermal parameters are taken into account (Section 4). Going forwards, we recommend the inclusion of temperature-dependent thermal parameters in future thermal models of subduction zones, especially when inferences on seismicity are made.

## 2   Methods

We base our models on the subduction zone community benchmark presented by van Keken et al. (2008). We use the tailor-made two-dimensional finite element Python code xFieldstone (Van Zelst, 2023, https://github.com/irisvanzelst/xFieldstone) to solve the incompressible Stokes equations with Crouzeix-Raviart (Crouzeix and Raviart, 1973) elements (also used in the MILAMIN code, (Dabrowski et al., 2008)) and the conservation of energy using quadratic triangular elements. xFieldstone is based on Fieldstone #68 which is part of the open source Fieldstone collection of educational finite element codes in compu-

tational geodynamics (https://cedrict.github.io/). The exact version of xFieldstone used to produce the results presented in this work can be found in Van Zelst (2023).

In the following, we first discuss the governing equations (Section 2.1) and rheology (Section 2.2) of the physical model. We then present the model setup (Section 2.3), our formulation for the thermal structure of the oceanic plate at the trench on the left-side of the model (Section 2.4), and the different functions we consider for the temperature-dependence of the thermal parameters (Section 2.5). Based on these functions, we define the parameter space of this study (Section 2.6) and detail the model diagnostics used in this work (Section 2.7).

## 2.1 Governing equations

Following van Keken et al. (2008), we solve the incompressible formulation of the conservation of mass and momentum (i.e., the Stokes equations) for velocity $v$ and pressure $p$:

$$\boldsymbol{\nabla} \bullet \boldsymbol{v} = 0, \tag{1}$$

$$\boldsymbol{\nabla} \bullet \boldsymbol{\sigma}' - \boldsymbol{\nabla} p = \boldsymbol{0}, \tag{2}$$

where $\boldsymbol{\sigma}'$ is the deviatoric stress tensor. In this formulation of the Stokes equations, we implicitly assume zero gravitational acceleration such that we have purely kinematically-driven subduction. This allows us to have the same (fixed) subduction geometry in all models.

We also solve for temperature $T$ using the steady-state conservation of energy without external heat sources such as radiogenic or shear heating to simplify the model:

$$\rho C_p(\boldsymbol{v} \bullet \boldsymbol{\nabla} T) - \boldsymbol{\nabla} \bullet (k\boldsymbol{\nabla} T) = 0, \tag{3}$$

where $\rho$ is density, $C_p$ is the heat capacity, and $k$ is the thermal conductivity. Unlike van Keken et al. (2008), we make these thermal parameters temperature-dependent instead of constant, as described in Section 2.5.

## 2.2 Rheology

We consider a purely viscous rheology, where we relate the deviatoric stress tensor $\boldsymbol{\sigma}'$ to the viscosity $\eta$ and the deviatoric strain-rate tensor $\dot{\varepsilon}$ according to:

$$\boldsymbol{\sigma}' = 2\eta\dot{\varepsilon}'. \tag{4}$$

Here, the deviatoric strain-rate tensor $\dot{\varepsilon}$ is defined as

$$\dot{\varepsilon}' = \frac{1}{2}\left(\boldsymbol{\nabla v} + \boldsymbol{\nabla v}^T\right). \tag{5}$$

Initially, we run sets of models with different viscous rheologies to successfully reproduce the different benchmark cases presented in van Keken et al. (2008) (Section S1; Figures S1-S7). In the following, we confine ourselves to a rheology that combines the diffusion and dislocation creep mechanisms used in van Keken et al. (2008). We implement this temperature-dependent rheology through an effective viscosity $\eta_{\text{eff}}$.

For the diffusion creep rheology, we use the simplified diffusion creep viscosity formulation $\eta_{\text{diff}}$ for olivine, where we set the activation volume to zero as it multiplies with the full pressure whereas the code only computes the excess pressure, because we neglect buoyancy forces (Eq. 2). Additionally, we ignore any effects caused by hydration and grain-size dependence, resulting in the following rheology:

$$\eta_{\text{diff}} = A_{\text{diff}} \exp\left(\frac{E_{\text{diff}}}{RT}\right), \tag{6}$$

where $A_{\text{diff}}$ is a prefactor, $E_{\text{diff}}$ is the activation energy, and $R$ is the universal gas constant. Similarly, we use the following expression for a dislocation creep rheology:

$$\eta_{\text{disl}} = A_{\text{disl}} \exp\left(\frac{E_{\text{disl}}}{nRT}\right) \dot{\varepsilon}_{II}^{(1-n)/n}, \tag{7}$$

where $A_{\text{disl}}$ is a prefactor, $E_{\text{disl}}$ is the activation energy, $n$ is the power-law exponent and $\dot{\varepsilon}_{II} = \sqrt{\dot{\varepsilon}_{xx}^2 + \dot{\varepsilon}_{xy}^2}$ is the square root of the second invariant of the deviatoric strain rate tensor.

We combine these formulations for diffusion and dislocation creep into one rheology by assuming two viscous dampers in series (Schmeling et al., 2008):

$$\eta_{\text{comb}} = \frac{\eta_{\text{diff}} \cdot \eta_{\text{disl}}}{\eta_{\text{diff}} + \eta_{\text{disl}}} = \left(\frac{1}{\eta_{\text{diff}}} + \frac{1}{\eta_{\text{disl}}}\right)^{-1} \tag{8}$$

To avoid unrealistically high stresses, we limit the maximum viscosity in the model to $\eta_{\text{max}} = 10^{26}$ Pa s for both the diffusion and dislocation creep rheology, such that the effective viscosity $\eta_{\text{eff}}$ becomes

$$\eta_{\text{eff}} = \left(\frac{1}{\eta_{\text{comb}}} + \frac{1}{\eta_{\text{max}}}\right)^{-1}. \tag{9}$$

## 2.3 Model setup

We use the two-dimensional model setup of the community benchmark for subduction zone modelling presented by van Keken
et al. (2008) (Figure 1). This setup is a highly simplified representation of a subduction zone. Its simplicity and the fact that
this setup is well-defined and thoroughly studied as a benchmark allow us to isolate the effect of the temperature-dependent
thermal properties in this study. Indeed, the simplicity of the model geometry allows us to add complexities in other parts of
the model, i.e., in this case temperature-dependent thermal properties. The modelling philosophy of this study is best described
as 'generic' according to Van Zelst et al. (2022), as we do not aim to obtain realistic results directly comparable to any specific
subduction zone.

We consider a domain that is $L_x = 660$ km wide and $L_y = 600$ km deep with the origin of the coordinate system at the
lower left corner and the $y$-axis positive upwards. We discretise the domain by means of a structured triangular grid with a
uniform resolution of 2.5 km, resulting in $528 \times 480$ triangular elements. We define a simple slab geometry with a $45°$ dip
angle originating at the top left corner and a 50 km thick overriding plate at the top of the model. The remaining part of the
model is the mantle wedge. Our chosen resolution ensures that the computational grid aligns with the bottom of the overriding
plate and the wedge corner.

We fix the overriding plate by prescribing no slip (i.e., zero velocity in both the $x$- and $y$-direction) at its bottom boundary
with the mantle wedge. We define the plate kinematics such that the downgoing slab subducts with a constant velocity of
5 cm/year by prescribing this velocity at the top of the slab from the corner point at $x = 50$ km and $y = 550$ km to the bottom
of the domain. At the corner point itself, we prescribe zero velocity.

For the conservation of energy, we apply a constant $0°$C temperature boundary condition along the top of the model domain.
At the right-hand boundary, we apply a linear temperature gradient in the overriding plate from $T = 0°$C at the top to $1300°$C
at the bottom of the overriding plate at $y = 550$ km. Below that, incoming material (i.e., $v_x < 0$) is assigned the maximum
temperature in the model $T_{\max} = 1300°$C. At the left boundary, we apply either a half-space cooling model with a slab age
$t_s$ of 50 Myr and constant thermal parameters (as used in the benchmark of van Keken et al. (2008)), or a temperature profile
extracted at 50 Myr from a one-dimensional cooling plate model (following Richards et al. (2018), and discussed further in
Section 2.4). The initial temperature field is constant with $T = 0°$C.

We first solve the Stokes equations across the entire domain. As we are only interested in the velocity field in the mantle
wedge, we overwrite the resulting velocity solution in the slab and overriding plate by our boundary conditions, i.e., no slip
in the overriding plate and a constant subduction velocity of 5 cm/year in the slab. With the velocity solution determined, the
heat equation is solved next. We then iteratively solve the Stokes and heat equation until convergence is reached, i.e. when the
horizontal and vertical components of the velocity and the temperature compared to the previous iteration change less than a
given tolerance. We choose a relative tolerance of $10^{-5}$ in our model runs for both velocity and temperature, although we also
impose a maximum number of 50 iterations to limit the wall-time of the model. Tests show that employing a lower tolerance of
$10^{-3}$ (reached before 50 iterations) changes the model diagnostics from Section 2.7 by less than $1°$C and has no effect on the

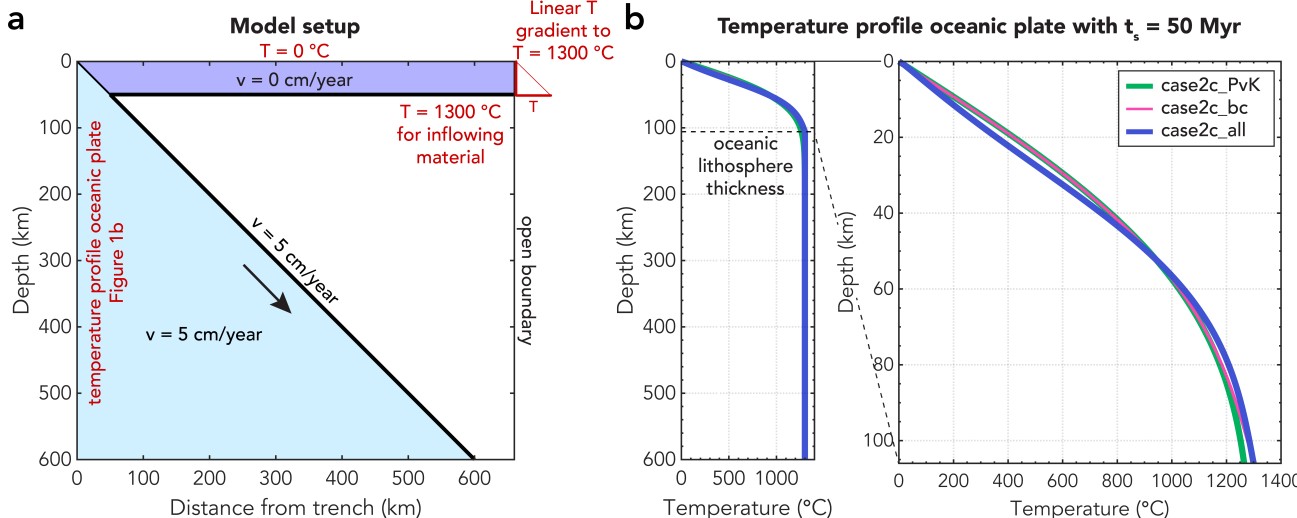

**Figure 1.** Model setup. (a) Model domain with kinematically prescribed overriding and subducting plate and temperature boundary conditions in red. Black bold lines indicate where we prescribe the velocities at the boundaries of the mantle wedge. (b) Different temperature boundary conditions for an oceanic plate with an age of 50 Myr used at the left-hand side of the model in (a) with a zoom of the top 106 km (i.e., the oceanic lithosphere thickness), below which the temperature is constant at $T = 1300°$C. The half-space model used by van Keken et al. (2008) is indicated in a thick green line (model case2c_PvK). We also show the two end-member plate models of our parameter study with the plate model with constant thermal parameters in pink (model case2c_bc) and the plate model considering temperature-dependence for all thermal parameters (model case2c_all) indicted in blue. See Figures S18 and S19 for the temperature profiles of all other models.

reported isotherm depths. To prevent numerical oscillations in the solution that inhibit convergence of the temperature field, we limit the change of each new iteration solution via a relaxation parameter $r$ after the first iteration according to

$$\phi_{\text{new}} = r \cdot \phi_{\text{new}} + (1 - r) \cdot \phi_{\text{old}} \tag{10}$$

where $\phi$ is the solution for $v_x$, $v_y$, and $T$. This relaxation step is applied to the velocity components after the Stokes solve and

to the temperature after the heat equation solve. After trial and error, we choose $r = 0.8$, which prevents numerical oscillations in the solution towards convergence.

## 2.4 Thermal structure of the oceanic plate at the trench

As the left-hand boundary condition for the conservation of energy, we prescribe the thermal structure of the incoming oceanic plate. In the original community benchmark, van Keken et al. (2008) used a simple half-space cooling model (Turcotte and Schubert, 2002) with a plate age of 50 Myr with constant values for the thermal conductivity, heat capacity, and density:

$$
T(y) = T_{\max} \cdot \mathrm{erf}\left( \frac{L_y - y}{2\sqrt{\frac{k}{\rho C_p} t_s}} \right). \tag{11}
$$

However, the half-space cooling model does not satisfy petrological constraints and fails to satisfy heat flow and bathymetric data for plate ages greater than $\sim 80$ Ma (Richards et al., 2018). Hence, we follow Richards et al. (2018) by including a more complex and realistic plate model as input for the temperature profile of the incoming oceanic plate. This plate model also has the advantage that it easily incorporates temperature-dependent thermal parameters, which results in consistency between the thermal structure of the incoming plate and the thermal structure we solve for in the rest of the domain.

We calculate the structure of the incoming oceanic plate in a linked, separate Python script, with the coordinate convention that the $y$-axis is positive downwards. The thermal structure of the oceanic plate is based on the one-dimensional heat equation (Turcotte and Schubert, 2002; McKenzie et al., 2005)

$$
\rho C_p \frac{\partial T}{\partial t} = \frac{\partial}{\partial y}\left( k \frac{\partial T}{\partial y} \right). \tag{12}
$$

Following McKenzie et al. (2005) and Richards et al. (2018), we discretise this equation using a one-dimensional time- and space-centered Crank-Nicolson finite difference scheme that is stable in both space and time and solve it numerically with a predictor-corrector step (Press et al., 1992) according to:

$$
-A\frac{k_{j-\frac{1}{2}}^m}{\Delta y_{j-1}^m} \cdot T_{j-1}^{n+1} + \left[ 1 + A\left( \frac{k_{j+\frac{1}{2}}^m}{\Delta y_j^m} + \frac{k_{j-\frac{1}{2}}^m}{\Delta y_{j-1}^m} \right) \right] \cdot T_j^{n+1} - A\frac{k_{j+\frac{1}{2}}^m}{\Delta y_j^m} \cdot T_{j+1}^{n+1} =
$$
$$
A\frac{k_{j-\frac{1}{2}}^m}{\Delta y_{j-1}^m} \cdot T_{j-1}^n + \left[ 1 - A\left( \frac{k_{j+\frac{1}{2}}^m}{\Delta y_j^m} + \frac{k_{j-\frac{1}{2}}^m}{\Delta y_{j-1}^m} \right) \right] \cdot T_j^n + A\frac{k_{j+\frac{1}{2}}^m}{\Delta y_j^m} \cdot T_{j+1}^n + B, \tag{13}
$$

with

$$
A = \frac{\Delta t}{\rho_j^m C_{p,j}^m \left( \Delta y_j^m + \Delta y_{j-1}^m \right)}, \tag{14}
$$

where $m = n$ for the predictor step and $m = n + \frac{1}{2}$ for the corrector step. Additionally, we have

$$B = -\frac{T_j^n \left( \rho_j^n C_{p,j}^n - \rho_j^{n-1} C_{p,j}^{n-1} \right)}{\rho_j^n C_{p,j}^n} \tag{15}$$

for the predictor step, and

$$B = -\frac{\left( T_j^{n+1} + T_j^n \right) \left( \rho_j^{n+1} C_{p,j}^{n+1} - \rho_j^n C_{p,j}^n \right)}{\rho_j^{n+1} C_{p,j}^{n+1} + \rho_j^n C_{p,j}^n} \tag{16}$$

for the corrector step.

As input parameters, we choose a constant $\Delta z$ of 1000 m and a constant time step $\Delta t = 1000$ year. We have the same temperature boundary conditions as the 2D model domain for consistency, with a surface temperature of $0°C$ and a maximum temperature (mantle potential temperature) of $1300°C$. We choose a plate thickness of 106 km in accordance with the optimum plate thickness found by Parsons and Sclater (1977); Sclater et al. (1980); McKenzie et al. (2005) based on heat flow observations. We recognise that this plate thickness diverges from the results of Richards et al. (2018), but their result involved the inclusions of compositional variability, in addition to the thermal dependence of material properties, which we do not include here. Hence, we use the older plate thickness value determination of McKenzie et al. (2005).

We solve for the temperature evolution of the incoming oceanic plate with the desired thermal parameters (Section 2.5) for 200 Myr, which we store in a lookup table (Figures S8-S18). The main part of the code then extracts the relevant temperature profile for a plate age of 50 Myr (van Keken et al., 2008) as input for the left boundary of the model domain, taking into account the different coordinate system conventions and the cubic interpolation between the 1D finite difference coordinates and finite element nodes in case of differing resolutions. We then solve the entire system using tridiagonal elimination.

## 2.5 Temperature-dependent thermal parameters

We use temperature-dependent expressions for the thermal conductivity, heat capacity, and density, using parameterisations based on observational experimental data for the way in which these values change with temperature.

Following McKenzie et al. (2005), we approximate the analytical expression for temperature-dependent thermal conductivity $k$ (Figure 2a) by Hofmeister (1999) with

$$k_H(T) = \frac{b}{1 + cT} + \sum_{m=0}^{3} d_m (T + 273)^m, \tag{17}$$

where $k$ has units of W m$^{-1}$ K$^{-1}$, although $T$ is in °C in this expression, and $b = 5.3$ W m$^{-1}$, $c = 0.0015$, $d_0 = 1.753 \cdot 10^{-2}$ W m$^{-1}$ K$^{-1}$, $d_1 = -1.0365 \cdot 10^{-4}$ W m$^{-1}$ K$^{-2}$, $d_2 = 2.2451 \cdot 10^{-7}$ W m$^{-1}$ K$^{-3}$, and $d_3 = 3.4071 \cdot 10^{-11}$ W m$^{-1}$ K$^{-4}$ are constants. This expression considers both heat transport and the radiative heat transfer by phonons.

Like McKenzie et al. (2005), we also implement the temperature-dependent conductivity for olivine proposed by Xu et al. (2004) to account for the large uncertainties in the temperature-dependence of the thermal conductivity:

$$k_X(T) = k_{298} \left( \frac{298}{T + 273} \right)^n,$$
(18)

where $T$ is in °C, $k_{298} = 4.08$ W m$^{-1}$ K$^{-1}$ and $n = 0.406$.

For the heat capacity $C_p$ (Figure 2b), we follow Berman (1988) to calculate the heat capacity of both fayalite and forsterite

(McKenzie et al., 2005) such that we have

$$C_{p,\text{fa}|\text{fo}} = \left( a_{0,\text{fa}|\text{fo}} + a_{1,\text{fa}|\text{fo}} \cdot T^{-\frac{1}{2}} + a_{3,\text{fa}|\text{fo}} \cdot T^{-3} \right) \cdot \frac{1000}{m_{\text{fa}|\text{fo}}},$$
(19)

where $C_p$ is the heat capacity in J kg$^{-1}$ K$^{-1}$ and $T$ is in K. We use updated values for the constants according to Berman and Aranovich (1996), resulting in $a_{0,\text{fa}} = 252$ J mol$^{-1}$ K$^{-1}$, $a_{1,\text{fa}} = -20.137 \cdot 10^2$ J mol$^{-1}$ K$^{-\frac{1}{2}}$, and $a_{3,\text{fa}} = -6.219 \cdot 10^7$ J K$^2$ mol$^{-1}$ for fayalite and $a_{0,\text{fo}} = 233.18$ J mol$^{-1}$ K$^{-1}$, $a_{1,\text{fo}} = -18.016 \cdot 10^2$ J mol$^{-1}$ K$^{-\frac{1}{2}}$, and $a_{3,\text{fo}} = -26.794 \cdot$

$10^7$ J K$^2$ mol$^{-1}$ for forsterite. To obtain the heat capacity in the correct unit of J kg$^{-1}$ K$^{-1}$, we multiply the equation from Berman (1988) where $C_p$ is in J mol$^{-1}$ K$^{-1}$ with $\frac{1000}{m_{\text{fa}|\text{fo}}}$, where $m_{\text{fa}|\text{fo}}$ is the molecular mass of fayalite (fa) or forsterite (fo). We then obtain the effective heat capacity in the model by assuming a molar fraction $f = 0.11$ of fayalite in the mantle according to McKenzie et al. (2005):

$$C_{p,\text{eff}} = (1 - f) \cdot C_{p,\text{fo}} + f \cdot C_{p,\text{fa}}.$$
(20)

For the dependency of density on temperature (Figure 2c), we follow the parameterisation of McKenzie et al. (2005) based on the integration of the parameterisation of the temperature-dependence of the thermal expansivity according to Bouhifd et al. (1996):

$$\rho = \rho_0 \exp \left( - \left[ \alpha_0 (T - T_0) + \frac{\alpha_1}{2} (T^2 - T_0^2) \right] \right),$$
(21)

where $T$ is in K, $\rho_0 = 3330$ kg m$^{-3}$, $T_0 = 273.15$ K, $\alpha_0 = 2.832 \cdot 10^{-5}$ K$^{-1}$, and $\alpha_1 = 3.79 \cdot 10^{-8}$ K$^{-2}$.

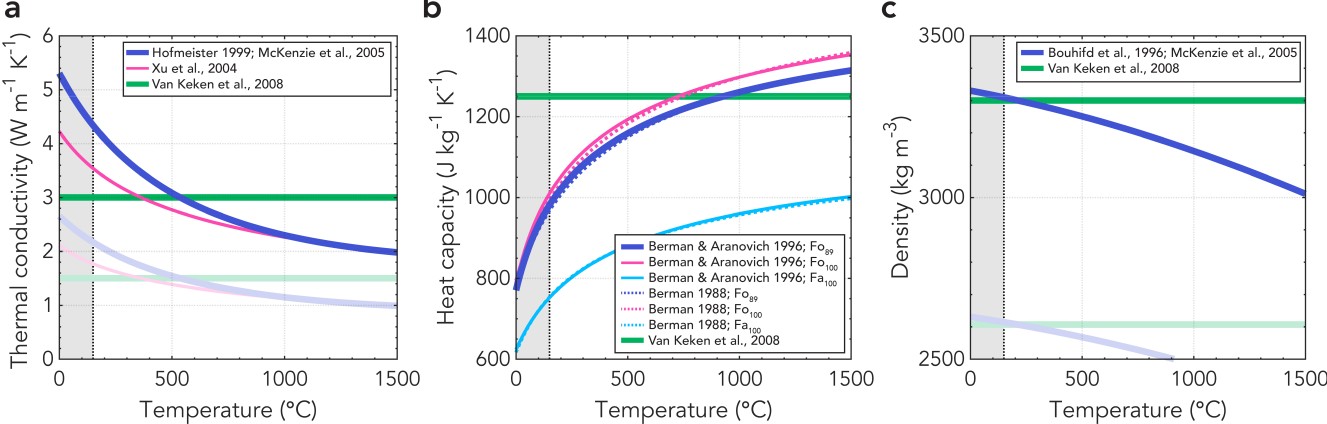

**Figure 2.** Temperature-dependence of (a) the thermal conductivity $k$ according to Xu et al. (2004) and the approximation of Hofmeister (1999) according to McKenzie et al. (2005); (b) the heat capacity $C_p$ according to Berman and Aranovich (1996) (solid lines) and Berman (1988) (dotted lines) for different ratios of forsterite (fo) and fayalite (fa); (c) the density according to the parameterisation by McKenzie et al. (2005) of Bouhifd et al. (1996). Constant values taken from van Keken et al. (2008) are plotted as reference in thick green lines. The lighter colours indicate the crustal approximation for the thermal conductivity (i.e., multiplied by 0.5) and the density (i.e., multiplied by 0.79). These crustal approximations are most relevant for rocks at temperatures lower than $150°C$ (indicated by the grey area and the vertical dotted line). Barring these approximations, the functions for the thermal parameters in this figure are relevant for mantle rocks at larger temperatures.

Other formulations for the temperature-dependence of the thermal conductivity, heat capacity, and density than the ones described here are also available (e.g., Berman and Brown, 1985; Seipold, 1998; Wen et al., 2015; Su et al., 2018), but here we limit ourselves to the formulations described in this section to test the effect of such variability.

     In our preferred formulations for the temperature-dependence of the thermal parameters, the thermal conductivity (Hofmeister, 1999; McKenzie et al., 2005) varies with a factor of 2.5 for the temperature range in our subduction zone models (i.e.,

$0 - 1300°C$) with $k = 5.3$ W m$^{-1}$ K$^{-1}$ for $T = 0°C$ and $k = 2.1$ W m$^{-1}$ K$^{-1}$ for $T = 1300°C$ (Figure 2). Similarly, the heat capacity (89% forsterite; Berman and Aranovich, 1996) varies by a factor of 1.65 over the temperature range $0 - 1300°C$. The temperature-dependence of the density (Bouhifd et al., 1996) is less pronounced, with a variation of approximately 9% in density for temperatures common to subduction zones. However, the formulations used here for the thermal conductivity, heat capacity, and density are mainly applicable to mantle rocks, which do not typically experience low temperatures of $T < 150°C$.

Instead, crustal rocks with different thermal properties are a better approximation at these temperatures, which is addressed further in the next section.

### 2.6   Parameter space

To systematically test the effect of temperature-dependent parameters on the thermal structure of subduction zones, we run the suite of simulations outlined in Table 1. We start with a reference model case2c_PvK based on the van Keken et al. (2008)

benchmark models. Note that this model is not in the original suite of benchmark models of van Keken et al. (2008); the difference being that the rheology employed is a combination of diffusion and dislocation creep. Then, we first test the effect

of adding the more complex temperature-profile of the plate model instead of the half-space cooling model in case2c_bc. We test the effect of the two different functions for the thermal conductivity with models case2c_k1 and case2c_k2, where the approximation of the function by Hofmeister (1999) is our preferred function, following McKenzie et al. (2005). Our preferred model for the heat capacity is the one where we use the function of Berman (1988) with updated values of Berman and Aranovich (1996) for a composition of 89% of forsterite and 11% fayalite (case2c_Cp6). We also test the effect of using the older values of Berman (1988) (case2c_Cp3), and a composition of 100% forsterite (case2c_Cp4) and 100% fayalite (case2c_Cp5). Here, the numbers behind $k$ and $C_p$ in the model names refer to the flags used in the code to select different options for the temperature-dependent thermal parameters. We test the temperature-dependent density in case2c_rho. Finally, we combine our preferred functions of the thermal parameters in simulation case2c_all.

To illustrate how the different simulations differ in terms of temperature-dependence of the thermal parameters, we show the thermal diffusivity $\kappa$ in Figure 3 calculated according to

$$\kappa = \frac{k}{\rho C_p}. \tag{22}$$

Hence, when all thermal parameters $k$, $C_p$, and $\rho$ are temperature-dependent in model case2c_all the overall temperature-dependency of the model is largest. Compared to the constant thermal diffusivity used in the benchmark by van Keken et al. (2008) values are up to 319% larger and up to 28% smaller for the temperature range of our model. Large values for the diffusivity translate to rapid heat transfer, meaning that cold regions heat up faster and hot regions cool down faster. In general, Figure 3 shows that the thermal diffusivity is higher for low temperatures, meaning that the cold top of the slab will be heated faster.

The top of the oceanic plate, where temperatures are low, and hence where thermal parameters differ most from the constant values used in van Keken et al. (2008), is the area of the plate where our assumption of a uniform mantle composition is most likely to be inappropriate. Crustal materials also have temperature-dependent thermal properties, but, with different mineralogical compositions to the olivine-dominated mantle, they can have quite different values, with conductivities for crustal materials at temperatures $< 150°$C typically being lower than those of mantle rocks (Figure 2; e.g., Grose and Afonso, 2013; Richards et al., 2018). Given the effect that such variation would have on the rate of temperature change in the top few kilometres of the slab, and the potential for this to insulate the deeper sections of the slab from the effects of the mantle wedge, we test the impact that incorporating a crustal layer in the subducting slab in our models might have using a simple parameterisation (see Sections 4.2 and 4.3 for a discussion on this simplification and alternatives). In order to approximate an oceanic crustal layer in the slab in our models, we define a crustal thickness of 7 km for which we use half the value of the thermal conductivity for a given temperature (e.g., Grose and Afonso, 2013, ; Figure 2). We then run our main set of 10 models with the addition of this oceanic crust parameterisation in the slab to assess how the inclusion of explicitly modelled crustal heterogeneity might affect the results presented in this work. We indicate this set of models with the postfix _mc (mimic crust) to our simulation names.

Another important aspect in a subduction zone is the compositional structure of the overriding plate. In our standard set of models there is no crustal layer within the overriding plate. To assess the effect of the crust of the overriding plate on the thermal structure of the slab, we run another two sets of models based on our main set of 10 models and including the oceanic crust parameterisation in the slab of the _mc models, but with an additional crustal layer in the overriding plate. The first set of models (denoted _op; oceanic plate) incorporates a crustal layer with a thickness of 7 km and half the thermal conductivity values for a given temperature, similar to our parameterisation of oceanic crust in the slab, as if the overriding plate were oceanic in origin. The second set of models considers a continental upper plate (denoted _cp; continental plate) with a crustal thickness of 35 km, halved thermal conductivity values, and a density that is multiplied by 0.79 to obtain realistic values for crustal densities at low temperatures.

Note that for these 3 sets of models that include a parameterisation for oceanic crust in the slab, we also include this crustal layer approximation in the 1-dimensional plate model that calculates the left temperature boundary condition. In contrast, the halfspace cooling model used by van Keken et al. (2008) does not allow for the addition of layers. Therefore, the boundary condition in these case2c_PvK models is inconsistent due to the limitations of the halfspace cooling model deployed by van Keken et al. (2008).

To illustrate the applicability of our results to the variety of subduction zones observed on Earth, we also run two end-member models with constant and temperature-dependent thermal parameters for a model with a younger ($t_s = 20$ Ma) and older ($t_s = 80$ Ma) slab age, compared to our reference slab age of $t_s = 50$ Ma.

## 2.7 Model diagnostics

To assess our models and quantify their differences, we use the three diagnostics defined in the community benchmark by van Keken et al. (2008), as well as the maximum depth of certain isotherms and the surface heat flux. Following van Keken et al. (2008), we define a uniform rectangular grid of $111 \times 110$ points with 6 km spacing starting in the top left corner and stored row-wise. On this grid, we interpolate the discrete temperature field $T_{ij}$ in °C in a postprocessing step. Using this grid, we output (1) the temperature $T_{x=60\text{km}}$ at the top of the slab at $y = 540$ km and $x = 60$ km, just downdip of the mantle wedge corner; (2) the $l^2$-norm of the temperature along the top of the slab $T_{\text{slab}}$ between $y = 600$ km and $y = 390$ km as defined by

$$T_{\text{slab}} = \sqrt{\frac{\sum_{i=1}^{36} T_{ii}^2}{36}}; \tag{23}$$

and (3) the $l^2$-norm of the temperature in the tip of the mantle wedge between 54 and 120 km depth as defined by

$$T_{\text{wedge}} = \sqrt{\frac{\sum_{i=10}^{21} \sum_{j=10}^{i} T_{ij}^2}{78}}. \tag{24}$$

**Table 1.** Simulations[a]

| Model[a] | $T$-profile left boundary | $k$ | $C_p$ | $\rho$ | Figures |
|---|---|---|---|---|---|
| case2c_PvK | half-space cooling model $t_s = 50$ Myrs | constant[b] | constant | constant | Fig. 4, S29 |
| case2c_bc | plate model $t_s = 50$ Myrs | constant | constant | constant | Fig. S20, S30 |
| case2c_k1 | plate model $t_s = 50$ Myrs | McKenzie et al. (2005) approximation of Hofmeister (1999) | constant | constant | Fig. S21, S31 |
| case2c_k2 | plate model $t_s = 50$ Myrs | Xu et al. (2004) | constant | constant | Fig. S22, S32 |
| case2c_Cp6 | plate model $t_s = 50$ Myrs | constant | 89% forsterite with values from Berman and Aranovich (1996) | constant | Fig. S23, S33 |
| case2c_Cp3 | plate model $t_s = 50$ Myrs | constant | 89% forsterite with values from Berman (1988) | constant | Fig. S24, S34 |
| case2c_Cp4 | plate model $t_s = 50$ Myrs | constant | 100% forsterite with values from Berman and Aranovich (1996) | constant | Fig. S25, S35 |
| case2c_Cp5 | plate model $t_s = 50$ Myrs | constant | 100% fayalite with values from Berman and Aranovich (1996) | constant | Fig. S26, S36 |
| case2c_rho | plate model $t_s = 50$ Myrs | constant | constant | McKenzie et al. (2005) parameterisation of Bouhifd et al. (1996) | Fig. S27, S37 |
| case2c_all | plate model | McKenzie et al. (2005) approximation of Hofmeister (1999) | 89% forsterite with values from Berman and Aranovich (1996) | McKenzie et al. (2005) parameterisation of Bouhifd et al. (1996) | Fig. 5, S38 |
| case2c_20PvK | half-space cooling model $t_s = 20$ Myrs | constant | constant | constant | Fig. 8 |
| case2c_20all | plate model $t_s = 20$ Myrs | McKenzie et al. (2005) approximation of Hofmeister (1999) | 89% forsterite with values from Berman and Aranovich (1996) | McKenzie et al. (2005) parameterisation of Bouhifd et al. (1996) | Fig. 8 |
| case2c_80PvK | half-space cooling model $t_s = 80$ Myrs | constant | constant | constant | Fig. 8 |
| case2c_80all | plate model $t_s = 80$ Myrs | McKenzie et al. (2005) approximation of Hofmeister (1999) | 89% forsterite with values from Berman and Aranovich (1996) | McKenzie et al. (2005) parameterisation of Bouhifd et al. (1996) | Fig. 8 |

[a] None of the models listed in this table include the oceanic crust parameterisation in the slab or an overriding plate. Instead, the first 10 models listed here also represent three additional batches of models. Models with the addition _mc (mimic crust) include the oceanic crust parameterisation in the slab without an overriding plate. Models with the addition _op (oceanic plate) include the oceanic crust parameterisation in the slab and a parameterisation for an oceanic upper plate. Models with the addition _cp (continental plate) also include the oceanic crust parameterisation in the slab and a continental overriding plate parameterisation. So, for example, model case2c_PvK does not include any oceanic crust parameterisation or overriding plate, while model case2c_PvK_cp does include the oceanic crust parameterisation in the slab and a parameterisation for a continental upper plate. [b] The constant values used for the thermal conductivity $k$, heat capacity $C_p$, and density $\rho$ are taken from van Keken et al. (2008) to be $k = 3$ W m$^{-1}$ K$^{-1}$, $C_p = 1250$ J kg$^{-1}$ K$^{-1}$, and $\rho = 3300$ kg m$^{-3}$.

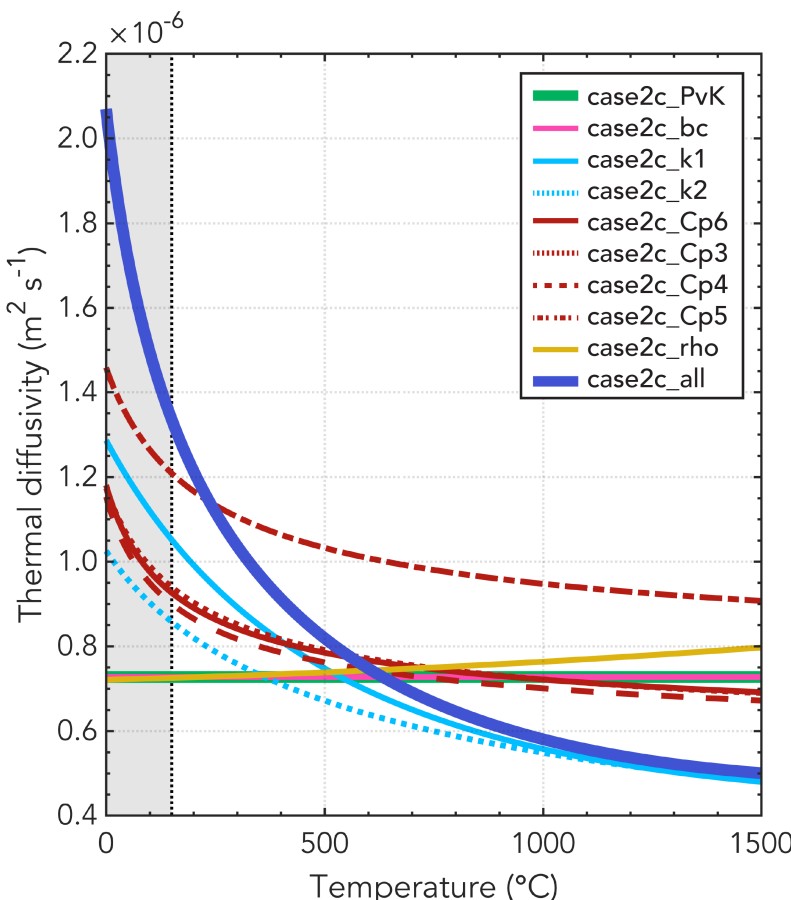

**Figure 3.** The thermal diffusivity $\kappa$ for our 10 main simulations (Table 1). The thermal diffusivity illustrates the overall temperature-dependence of the model by combining the thermal conductivity $k$, heat capacity $C_p$, and density $\rho$ according to $\kappa = \frac{k}{\rho C_p}$. The derived functions of thermal diffusivity for our simulations are relevant for mantle rocks at temperatures typically larger than $150°C$ (indicated by the vertical dotted line). Therefore, the temperature range for which these functions are not a good representation is indicated by the grey area.

In addition to the diagnostics previously used in van Keken et al. (2008), we further report additional diagnostics that relate more closely to changes in the thermal structure of the slab that impact other processes, particularly the main processes governing seismogenesis. We report the maximum depth of the $350°C$ and $450°C$ isotherms within the slab, which are associated with the brittle-ductile transition and hence the downdip limit of the seismogenic zone of megathrust seismicity (Hyndman et al., 1997; Gutscher and Peacock, 2003). We also report the maximum depth of the $600°C$ isotherm in the slab, which, together with the $350°C$ and $450°C$ isotherms, is associated with the main dehydration reaction fronts within the slab, and the associated intermediate-depth seismicity between 70 km and 350 km depth (Peacock, 2001; Yamasaki and Seno, 2003; Kelemen and Hirth, 2007). In addition, we report the values of the surface heat flux in our models, since the observed surface heat flux

is a frequently-used constraint for models. Besides that, we provide snapshots of relevant variables, such as the temperature, viscosity, and velocity.

Postprocessing and visualisation is primarily done using Matlab scripts (available in the Zenodo directory) with additional touch-ups in Adobe Illustrator. We use scientific colour maps by Crameri (2018a); Crameri et al. (2020) to avoid visual distortion of the data and exclusion of readers with colour-vision deficiencies (Crameri, 2018b). To compare the thermal parameters and initial temperature conditions of the different models, we colour the models according to the optimal qualitative colour palette by Anton Tsitsulin (2019; retrieved: May 10, 2021).

## 3  Results

### 3.1  Models with constant thermal parameters

The results from the reference model case2c_PvK with constant thermal parameters are shown in Figure 4. It shows a subducting plate with a relatively cold core and a cold overriding plate with the base of the overriding plate that spills into the mantle wedge. There is flow in the mantle wedge around the base of the overriding plate which reaches the tip of the mantle wedge at $x = 50$ km and $y = 550$ km and heats up the subducting plate from the top.

The reference model has a combined dislocation and diffusion creep rheology in contrast to the original cases presented in van Keken et al. (2008) which are either isoviscous (Figures S1-S4), purely diffusion creep (Figure S5), or purely dislocation creep (Figure S6). Despite the difference in rheology, the model diagnostics of our reference model do not change significantly with respect to the model with a pure dislocation or diffusion creep rheology presented in van Keken et al. (2008) (Figure S7). However, looking at the snapshots presented in Figure 4 and comparing them to the benchmark models of van Keken et al. (2008) (Figure S5,6), there are distinct differences between our reference model and the benchmark cases presented in van Keken et al. (2008) in terms of the viscosity and velocity field in the mantle wedge, as well as the temperature field within the slab. These differences are not evident from our quantitative model diagnostics, as the differences manifest themselves at high temperatures in the mantle wedge. These high temperatures and the region of the mantle wedge are not included in our model diagnostics, as they principally affect the area of the model domain outside the main focus of our study, i.e., the slab.

In model case2c_bc, we build upon our reference model and change the initial and boundary temperature condition of the subducting oceanic plate at the left of the model from a half-space cooling model to the plate model. This does not incur major changes in the model diagnostics (Table 2), consistent with the similarity between the temperature profiles of the half-space cooling model and the plate model with constant values for the thermal parameters (Figure 1b).

### 3.2  Models with temperature-dependent thermal conductivity

Using the temperature-dependent thermal conductivity according to Hofmeister (1999); McKenzie et al. (2005) in model case2c_k1 results in an overall colder model with the slab isotherms penetrating deeper into the mantle. This effect increases with temperature with the $350°C$ isotherm reaching 20 km deeper into the mantle and the $600°C$ isotherm reaching almost

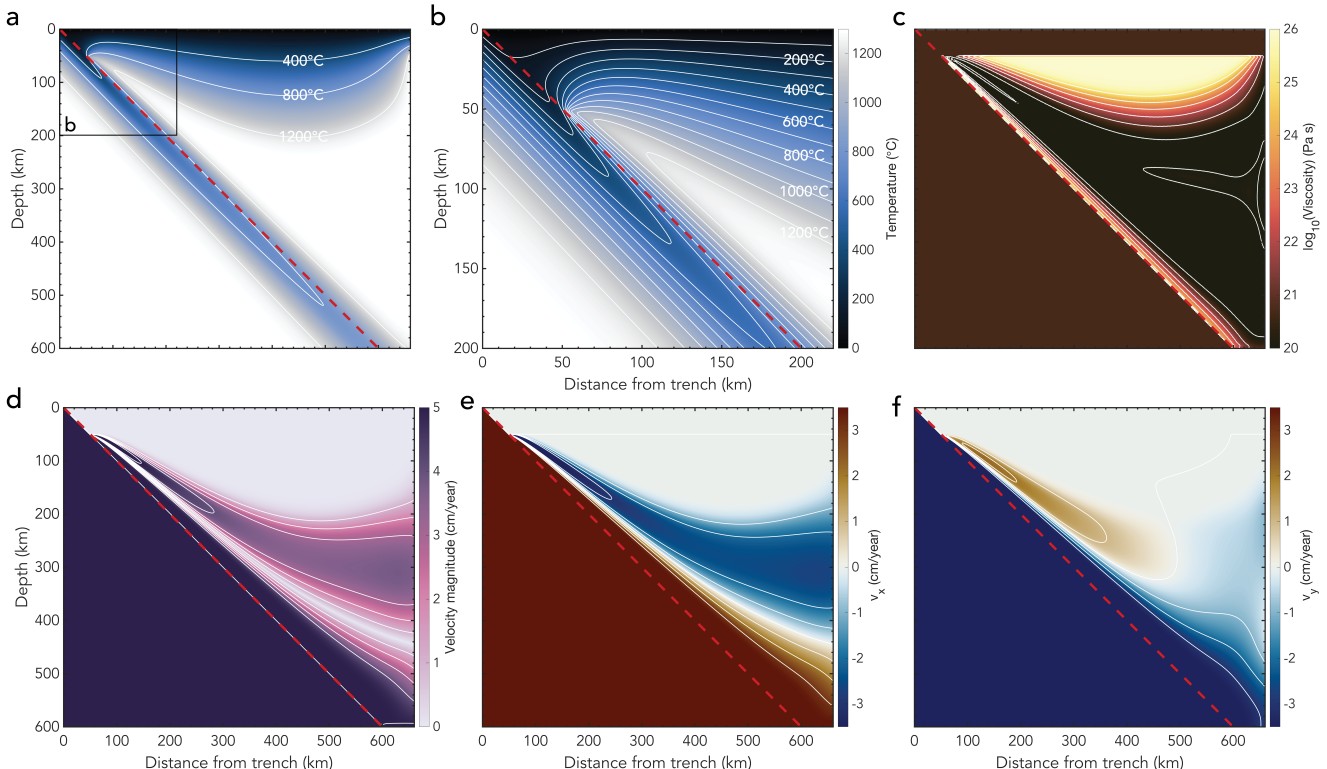

**Figure 4.** Snapshots of different variables for model case2c_PvK with constant values for the thermal parameters based on van Keken et al. (2008) including both a dislocation and diffusion creep rheology (Table 1). (a) Temperature field with isotherms indicated in white; (b) zoom of the temperature field; (c) viscosity with white contours for $\eta = 10^{20}$ Pa s to $\eta = 10^{26}$ Pa s with intervals of one order magnitude; (d) velocity magnitude with white contours for $v = 0$ cm/year to $v = 5$ cm/year with intervals of 1 cm/year; (e) horizontal component of the velocity with white contours for $v_x = -3$ cm/year to $v_x = 3$ cm/year with intervals of 1 cm/year; (f) vertical component of the velocity with white contours for $v_y = -3$ cm/year to $v_y = 3$ cm/year with intervals of 1 cm/year. The red dashed line indicates the top of the subducting slab.

90 km deeper into the mantle compared to the reference model (Figure 7). We observe a similar but less-pronounced trend when we use the thermal conductivity by Xu et al. (2004).

### 3.3 Models with temperature-dependent heat capacity

When using a temperature-dependent heat capacity, the model diagnostics show larger temperatures in the mantle wedge compared to the reference model with a constant heat capacity value. Similarly, the subducting slab is warmer and isotherms penetrate less deep into the mantle. For our preferred heat capacity model with 89% forsterite and values from Berman and Aranovich (1996), the temperature diagnostics in the mantle wedge are larger by up to 37.7°C and the maximum depths reached by the isotherms in the slab are shallower by 13.7 - 50 km (Figure 7).

Using the values of Berman (1988) instead of the updated values of Berman and Aranovich (1996) only incurs minor changes in the model diagnostics of maximum 0.9°C and 1.3 km. Changing the ratio of forsterite and fayalite to 100% forsterite in model

case2c_Cp4 results in a slightly warmer mantle wedge by up to 2.8°C and shallower slab isotherms identical to the isotherm

depths obtained in model case2c_Cp3 with values from Berman (1988) (Figure 7). In the purely fayalite model case2c_Cp5, the

heat capacity is lower, resulting in a model that is cooler than the model with 89% forsterite (case2c_Cp6), but still warmer than

the reference model with a constant value for the heat capacity. Disregarding the temperature-dependent aspect of heat capacity

tested, the overall magnitudes of the heat capacity used in the four $C_p$ models from Figure 2b also differs. For example, the

pure fayalite heat capacity model has the lowest overall heat capacity. This trend in changing magnitude of the heat capacity is

also consistently visible in the model results with models with lower heat capacity exhibiting lower temperatures and models

with higher heat capacity resulting in higher temperature diagnostics. However, it is not straightforward to include the model

with constant heat capacity values in this trend as well. For example, model case2c_Cp5 with fayalite values consistently

has a lower heat capacity than the reference model with constant values, but the overall model diagnostics still show larger

temperatures like the models with both larger and smaller heat capacity magnitudes depending on the temperature. Hence, the

temperature-dependence of the heat capacity has non-linear effects on the resulting temperature field.

### 3.4 Models with temperature-dependent density

When we use a temperature-dependent density in model case2c_rho the model is cooler than the reference model case2c_PvK,

but the effect is less pronounced than for the thermal conductivity (Table 2; Figure 7). This results in isotherms that reach

deeper into the mantle.

### 3.5 Models including all three temperature-dependent thermal parameters

We show the results for the model case2c_all in Figure 5. In this model, we include the temperature-dependent function for

the thermal conductivity by Hofmeister (1999); McKenzie et al. (2005), the function of the heat capacity for 89% forsterite

with values from Berman and Aranovich (1996), and the temperature-dependent density. We show the differences between this

model and the reference model (Figure 4) in Figure 6a-b for easy comparison. Based on our model diagnostics (Table 2), the

model is overall colder than the reference model and the slab has a colder core with isotherms that reach deeper into the mantle

when we use temperature-dependent expressions for all thermal parameters (Figure 7). However, the effect is less pronounced

than for the models where we only use a temperature-dependent expression for the thermal conductivity. This is likely because

the warming effect of the temperature-dependent heat capacity cancels part of the cooling effect of using temperature-dependent

thermal conductivity and density. The largest difference between the two models is the temperature structure of the overriding

plate, which is colder in the temperature-dependent model. Although we specifically focus on the change in thermal structure

in the slab in this work, the difference in temperature in the overriding plate due to the use of temperature-dependent thermal

parameters instead of constant values affects the thermal structure of the slab. Since the overriding plate is colder in models

including temperature-dependence of the thermal parameters, the heating of the interface between the slab and the overriding

plate is delayed, which likely plays an important role in time-dependent models of the thermal evolution of slab dynamics.

Within the slab itself, Figure 6a-b shows that the largest temperature differences are approximately −65°C in the shallow

part. The top of the slab is colder in model case2c_all, allowing isotherms to reach deeper into the mantle. The difference

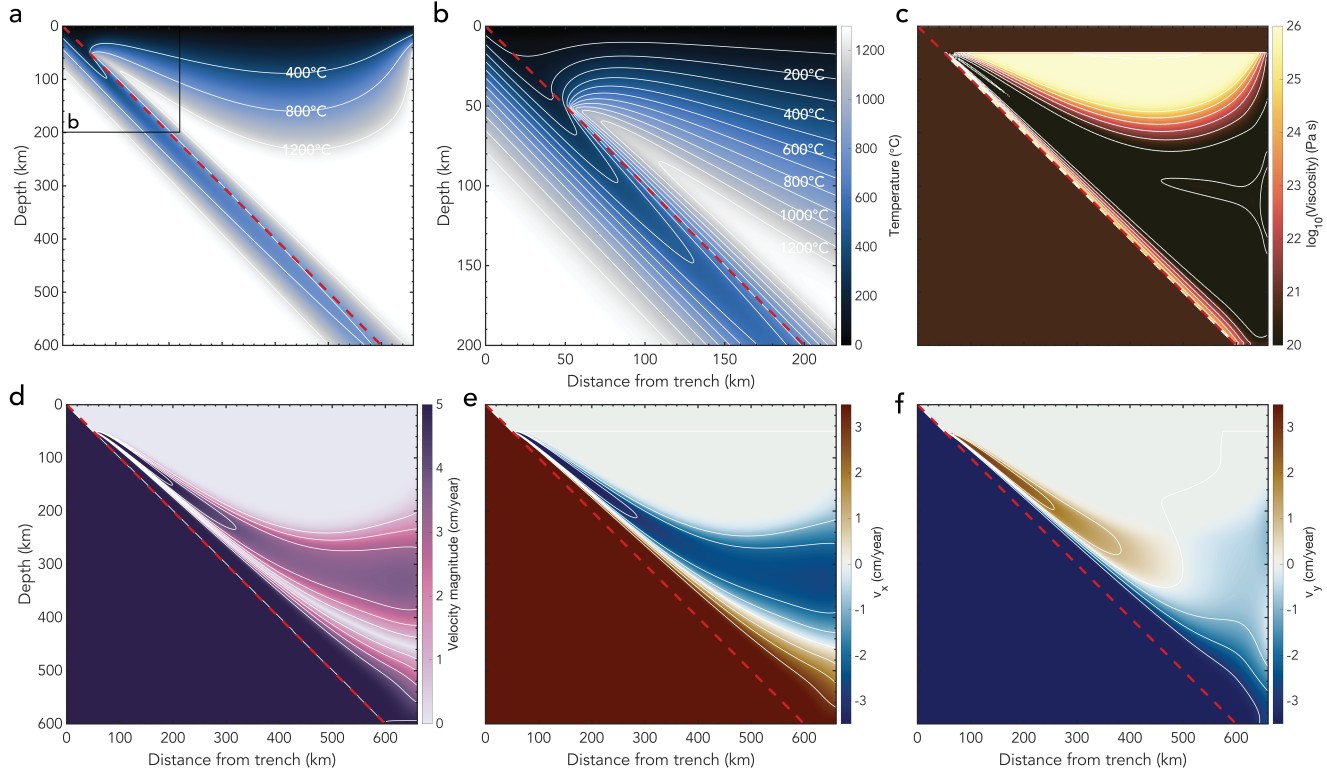

**Figure 5.** Snapshots of different variables for model case2c_all with our preferred temperature-dependent functions for all thermal parameters $k$, $C_p$, and $\rho$ (Table 1). (a) Temperature field with isotherms indicated in white; (b) zoom of the temperature field; (c) viscosity with white contours for $\eta = 10^{20}$ Pa s to $\eta = 10^{26}$ Pa s with intervals of one order magnitude; (d) velocity magnitude with white contours for $v = 0$ cm/year to $v = 5$ cm/year with intervals of 1 cm/year; (e) horizontal component of the velocity with white contours for $v_x = -3$ cm/year to $v_x = 3$ cm/year with intervals of 1 cm/year; (f) vertical component of the velocity with white contours for $v_y = -3$ cm/year to $v_y = 3$ cm/year with intervals of 1 cm/year. The red dashed line indicates the top of the subducting slab.

in isotherm depth is 3.8 km for the 350°C isotherm, 10 km for the 450°C isotherm, and 28.8 km for the 600°C isotherm (Figure 7). At the base of the lithosphere the bottom of the slab is warmer by up to 35°C compared to the reference model. This difference in slab temperature between the two models is partly due to the difference in boundary condition, as the plate cooling model in case2c_all is cooler than the half-space cooling model of case2c_PvK at shallow depths due to increased conductivity at low temperatures, and warmer at larger depth due to the imposition of a lower thermal boundary condition in the plate model. This also depends on the age of the subducting slab (see Section 3.6).

To summarise the effect of using temperature-dependent thermal parameters for all our models with a plate age of 50 Myr, we plot the maximum depth of the 350°C, 450°C, and 600°C isotherms for each model in Figure 7. With respect to the reference model with constant values, adding the temperature-dependent thermal conductivity by Hofmeister (1999); McKenzie et al. (2005) results in the largest changes in isotherm depth, with the isotherms reaching greater depths. Using a temperature-

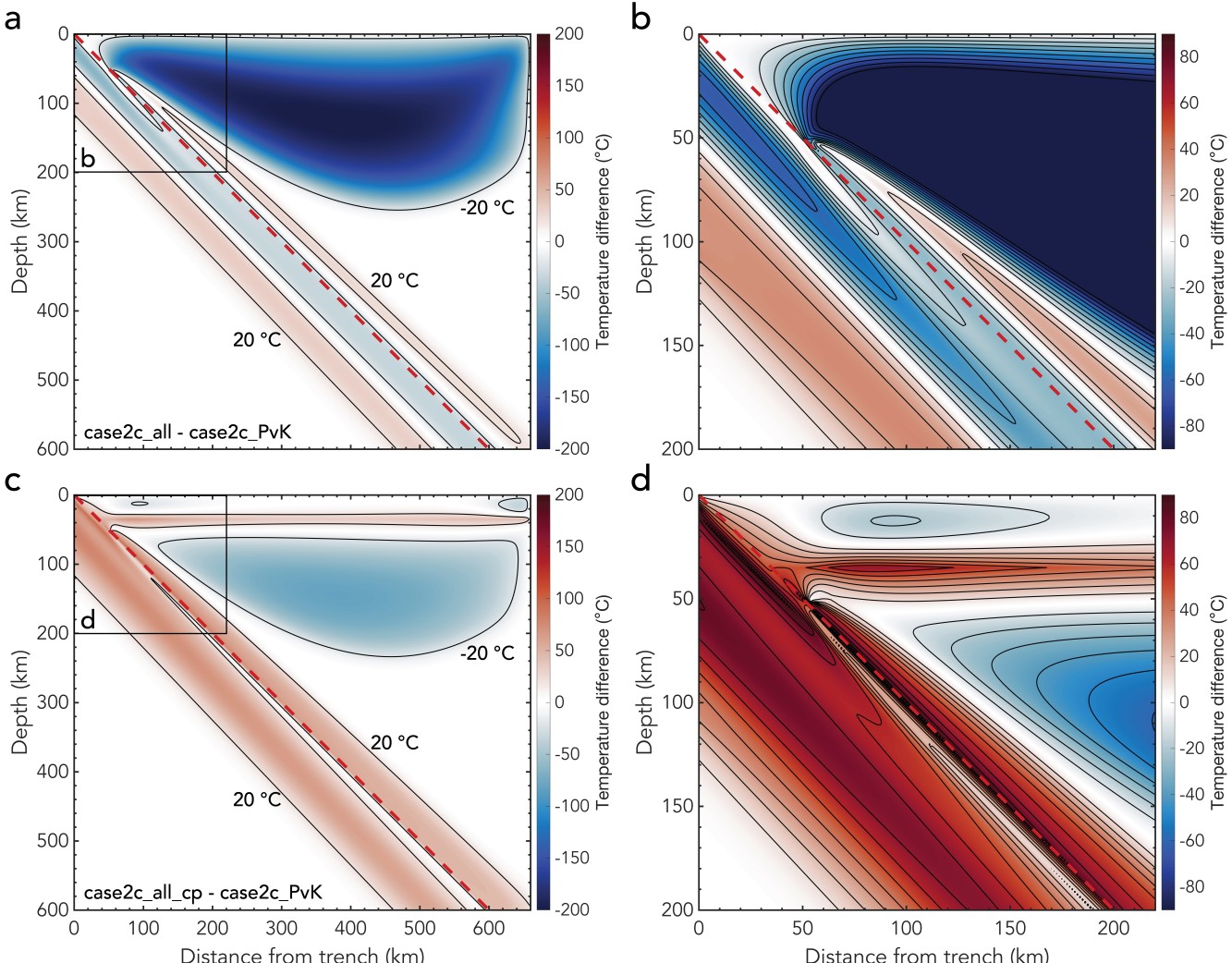

**Figure 6.** (a) Difference in temperature field between model case2c_all (Figure 5) and model case2c_PvK (Figure 4) with (b) a zoom of the top left corner of the model. (c) Difference in temperature field between model case2c_all_cp that includes the approximation for a crustal layer in the slab and a continental overriding plate and model case2c_PvK (Figure 4) with (d) a zoom of the top left corner of the model. Contours of the temperature difference are indicated in black. Note that panels (b and d) have a different colour scale than panels (a and c) to highlight the differences between the two models within the slab. The red dashed line indicates the top of the subducting slab.

dependent density also results in a colder top of the slab with deeper isotherms. In contrast, using a temperature-dependent

heat capacity yields a warmer slab with isotherms penetrating the mantle less deep than the reference model. Combining the

effect of temperature-dependent thermal conductivity, heat capacity, and density results in an overall effect of slab cooling with

the isotherms reaching greater depths.

The surface heat flux in the models varies across the surface with values of $\sim 17 - 49$ mW/m$^2$ observed in our models

(Figure S28). This is in line with typical surface heat flux values found by Davies and Davies (2010). At the boundaries of the

models unrealistically high surface heat fluxes are obtained as a result of artificial boundary effects. The surface heat fluxes of all models generally show the same trend and similar values. One exception to this is the heat flux observed in case2c_all which is 10 mW/m$^2$ larger near the trench than the heat flux observed in case2c_PvK and case2c_bc. Considering that typical surface heat flux values are in the range of tens of mW/m$^2$ (Davies and Davies, 2010), this is a significant change in model predicted surface heat flux. So, our results show that including temperature-dependent thermal parameters increases the predicted surface

heat flux.

**Table 2.** Model diagnostics*

| | $T_{x=60km}$ (°C) | $T_{slab}$ (°C) | $T_{wedge}$ (°C) | Max depth 350°C (km) | Max depth 450°C (km) | Max depth 600°C (km) |
|---|---|---|---|---|---|---|
| case2c_PvK | 578.5 | 604.9 | 999.5 | 77.5 | 110.0 | 203.8 |
| case2c_bc | 578.4 | 604.8 | 999.5 | 77.5 | 110.0 | 203.8 |
| case2c_k1 | 526.0 | 553.6 | 948.9 | 97.5 | 148.8 | 291.3 |
| case2c_k2 | 549.8 | 573.0 | 975.3 | 90.0 | 135.0 | 260.0 |
| case2c_Cp6 | 616.2 | 642.4 | 1007.6 | 63.8 | 87.5 | 153.8 |
| case2c_Cp3 | 616.9 | 643.3 | 1007.8 | 63.8 | 86.3 | 152.5 |
| case2c_Cp4 | 618.9 | 644.0 | 1010.4 | 63.8 | 86.3 | 152.5 |
| case2c_Cp5 | 588.4 | 626.1 | 979.0 | 66.3 | 91.3 | 161.3 |
| case2c_rho | 566.6 | 593.7 | 992.0 | 81.3 | 117.5 | 221.3 |
| case2c_all | 552.6 | 581.6 | 949.8 | 81.3 | 120.0 | 232.5 |
| case2c_20PvK | 631.8 | 670.6 | 1008.6 | 53.8 | 70.0 | 108.8 |
| case2c_20all | 602.9 | 643.9 | 957.9 | 56.3 | 73.8 | 121.3 |
| case2c_80PvK | 558.5 | 578.9 | 996.0 | 97.5 | 148.8 | 293.8 |
| case2c_80all | 533.6 | 556.9 | 946.6 | 102.5 | 162.5 | 333.8 |
| case2c_PvK_cp | 659.2 | 664.4 | 1056.6 | 85.0 | 123.8 | 228.8 |
| case2c_bc_cp | 703.0 | 708.6 | 1064.0 | 60.0 | 80.0 | 151.3 |
| case2c_k1_cp | 651.3 | 658.4 | 1027.1 | 73.8 | 112.5 | 223.8 |
| case2c_k2_cp | 674.1 | 678.3 | 1048.5 | 68.8 | 100.0 | 195.0 |
| case2c_Cp6_cp | 736.3 | 741.2 | 1071.5 | 50.0 | 66.3 | 112.5 |
| case2c_Cp3_cp | 737.1 | 742.1 | 1071.7 | 50.0 | 66.3 | 112.5 |
| case2c_Cp4_cp | 738.7 | 742.9 | 1073.9 | 50.0 | 66.3 | 112.5 |
| case2c_Cp5_cp | 710.7 | 724.1 | 1046.2 | 51.3 | 68.8 | 122.5 |
| case2c_rho_cp | 691.5 | 697.7 | 1057.0 | 62.5 | 86.3 | 165.0 |
| case2c_all_cp | 675.7 | 682.9 | 1028.1 | 62.5 | 90.0 | 177.5 |

*See Table S1 in the supplementary material for model diagnostics of the models including an oceanic crust approximation in the slab without an upper plate and including an oceanic upper plate

## 3.6 Models with different slab ages

Similar to the models with a plate age of 50 Myr, we see a cooling effect in the temperature-dependent thermal models for the models with different plate ages (Figure 8), with the changes to the thermal structure of the slab more pronounced with increasing slab age. Similarly, from Figure 9, we see that there is a particularly strong trend when it comes to larger isotherms

such as 600°C with the differences between the reference models including constant thermal parameters and the models with

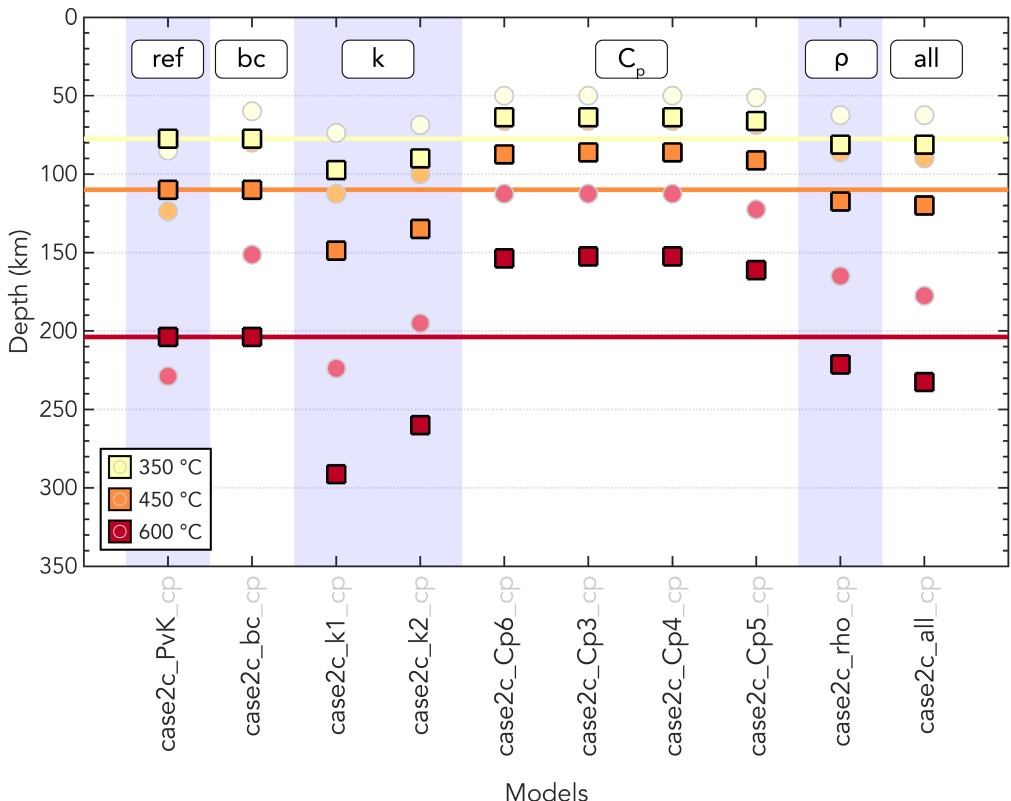

**Figure 7.** Change in maximum isotherm depth within the slab for models with different variations of temperature-dependent thermal parameters (Table 1). The three isotherm depths plotted here are the same as the ones from the model diagnostics in Table 2. Square symbols indicate models without the oceanic crust parameterisation in the slab and without an overriding plate. Faded circles and grey lines in the background indicate the _cp models including the oceanic crust parameterisation and the parameterisation for the continental upper plate. Different groups of models (i.e., testing different functions for the temperature-dependence of the thermal conductivity $k$) are indicated by vertical bands for clarity. Here, 'ref' refers to the reference model case2c_PvK. Horizontal coloured lines highlight the reference values of model case2c_PvK for easy comparison.

variable properties increasing when the plate gets older. Hence, including temperature-dependent thermal parameters has a larger effect for old, cold subducting plates. This is expected, as the functions used in this paper for temperature-dependent thermal properties (Figure 2) have their most extreme values at lower temperatures, which are more prevalent in older, and hence colder, slabs.

**3.7   Models with a parameterised oceanic crust in the slab**

When we include a parameterisation for a crustal layer at the top of the oceanic plate in the model, the diagnostics show a warmer top of the slab and mantle wedge. Within the slab temperatures are also warmer, resulting in shallower maximum

depths of the isotherms within the slab (Table S1; Figure S29; results are similar to the results of the overriding plate models shown in Figure 7, see next section).

The difference between isotherm depths for models with and without the oceanic crust parameterisation becomes more pronounced with increasing temperature. So, for example, the difference in depth of the $350°$C isotherm is 16.3 km, whereas it is 47.5 km for the depth of the $600°$C isotherm between models case2c_bc and case2c_bc_mc. Even though the isotherm depths are shallower and the wedge is warmer compared to the models without any oceanic crust parameterisation in the slab, the observed trends concerning the effects of temperature-dependent thermal parameters are the same as for the models without
an oceanic crust parameterisation in the slab (Figure S29).

An exception to these trends is the reference model case2c_PvK_mc which uses the half-space cooling model as a left-hand-side temperature boundary condition that does not account for the 7 km oceanic crust layer in the slab. Using this simplified boundary condition actually results in a colder slab with deeper isotherms (Figure S29). The large differences between the model case2c_PvK_mc and the other models in the _mc model batch show that the choice of temperature boundary condition
and the use of a consistent temperature boundary condition is crucial.

### 3.8    Models with a parameterised overriding plate

The models that include the parameterisation for an overriding plate also include the oceanic crust parameterisation in the slab. The models including an oceanic or continental plate all show the same trends as the models that do not include an overriding plate parameterisation, but do include the oceanic crust parameterisation. Typically, the models including a continental overriding plate result in a warmer slab and shallower isotherms compared to the other models. Figure 7 shows the results of
the models including a continental overriding plate, which are similar to the models with an oceanic overriding plate and the models without an overriding plate (Section 3.7; Figure S29). Indeed, the largest difference in isotherm depth is 8.75 km for the $600°$ isotherm between models case2c_k1_cp and case2c_k1_mc, but the results are predominantly similar enough such that they plot on top of each other (Figure S29). This indicates that the nature of the overriding plate, and indeed the inclusion of an overriding plate at all, does not significantly affect the temperature field in the subduction zone in our models and hence
does not affect the conclusions of this work. Beyond the slab, the inclusion of a continental overriding plate results in a warmer base of the continental crust and a colder mantle wedge (Figure 6c-d).

### 4    Discussion

We find that using temperature-dependent thermal parameters does not significantly change the first-order thermal structure
of a subduction zone in our models, with the large-scale features remaining the same. Hence, when considering large-scale subduction dynamics, the use of temperature-dependent thermal parameters is likely not an important factor. However, our results clearly show that the temperature-dependent thermal parameters do affect the thermal structure of the slab on the order of tens of degrees and hence tens of kilometres in these simple models of subduction zones. On a similar scale, our results

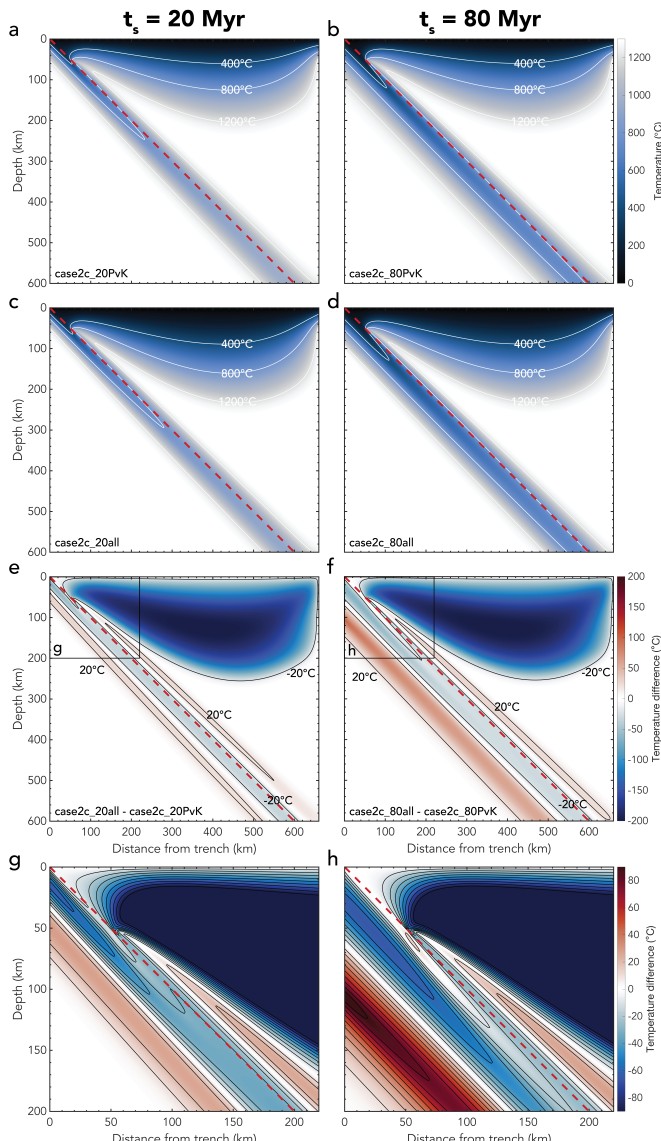

**Figure 8.** (a-d) Snapshots of the temperature field for models with a slab age of (a,c) 20 Myr and (b,d) 80 Myr with (a,b) constant and (c,d) variable thermal parameters (Table 1). (e-h) Difference in temperature field between (e,g) model case2c_20all and model case2c_20PvK with (g) a zoom of the top left corner of the model. (f,h) Same as panels (e,g) but for a slab age of 80 Myr. Contours of the temperature difference are indicated in black. In panels (g,h) contours for every 10° temperature difference are drawn. Note that panels (g,h) have a different colour scale than panel (e,f) to highlight the differences between the two models within the slab and to easily compare to Figure 6. The red dashed line indicates the top of the subducting slab.

show that the temperature boundary condition at the left-hand-side of the model influences the temperature field of the slab.

On the other hand, the inclusion of an overriding plate does not significantly affect the temperature field of the modelled slab.

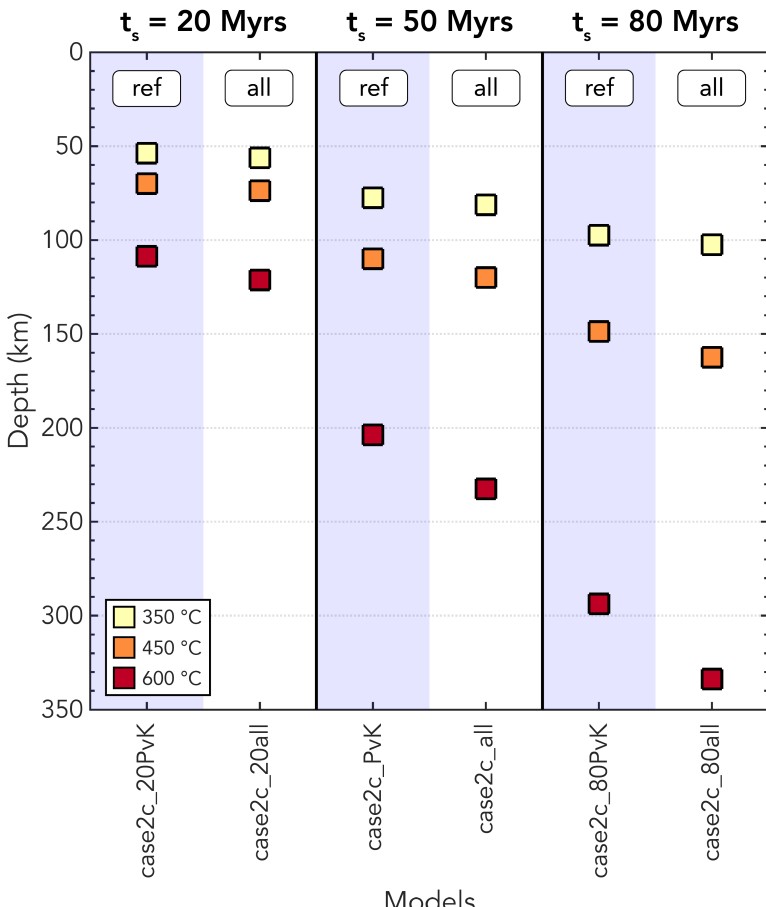

**Figure 9.** Change in maximum isotherm depth within the slab for end-member models with different subducting plate ages (Table 1). The three isotherm depths plotted here are the same as the ones from the model diagnostics in Table 2. The models with constant values are indicated by 'ref'.

Using temperature-dependent thermal parameters also affects the predicted surface heat flux in our models. As surface heat flux is one of the principal observables used in constraining subduction zone thermal models, all regional subduction zone thermal models are therefore affected by the inclusion of temperature-dependent thermal parameters. Our models with different plate ages show that our conclusions are valid regardless of the slab age, but they still lack realism in terms of model geometry 475 and the inclusion of many important processes relevant for the development of a realistic thermal structure of the slab.

The found effect of using temperature-dependent thermal parameters on the order of tens of degrees could be significant when assessing the detailed thermal structures of local slabs for seismological applications. In this section, we first discuss the implications of our results on modelling the thermal structure of subduction zones in light of megathrust, intermediate-depth, and deep seismicity, while taking into account the simple and geometrically-unrealistic nature of our models (Section 4.1). 480 As our models are conceptual calculations for the impact of including temperature-dependent variables, these implications are

generic, rather than applicable directly to any given subduction zone (Van Zelst et al., 2022). We further discuss the potential implications of our thermal models for the geochemical and mineralogical evolution of the slab, and the impact this may have on the flux of fluids through subduction zones (Section 4.2). We then discuss how realistic our models are, their limitations, and how future work could further improve both the models, and their applicability (Section 4.3).

## 4.1 Implications for seismicity

The temperature structure of a slab determines to a large extent where seismicity is expected to occur, through its effect on both the mode of failure and the onset of ductile behaviour, and its control on geochemical transitions within the slab and along the megathrust interface, including dehydration reactions. Here we summarise those effects and highlight how the results presented in Section 3 translate to influences on the distribution and extent of intermediate-depth and deep-focus earthquakes, and potentially on the extent of megathrust and related shallow seismicity.

### 4.1.1 Intermediate-depth seismicity

Although the shallow slab geometry in our model is clearly a simplification, at depths consistent with intermediate-depth seismicity, the slab dip of $45°$ in our models is realistic, with an average slab dip of $45.5°$ reported by Syracuse et al. (2010) in nature, although it remains highly variable between different subduction zones. Other studies also find that the slab dip is steeper away from the interface between the slab and the overriding plate (e.g., Jarrard, 1986; King, 2001; Hu and Gurnis, 2020). Therefore, we can make some inferences on the expected depth of intermediate-depth seismicity using our models. Intermediate-depth seismicity at depths of 75 - 300 km is commonly associated with a temperature range between $600°C$ and $800°C$, where dehydration embrittlement of the metamorphosed minerals in the slab occurs (e.g., Jung et al., 2004; Wang et al., 2017). Focusing on the $600°C$ isotherm in our models (Table 2; Figure 7), we see that its depth changes significantly throughout our parameter space with a depth of 203.8 km for the reference model case2c_PvK, 232.5 km for model case2c_all, and end members of 291.3 km depth for model case2c_k1 and 152.5 km depth for models case2c_Cp3 and case2c_Cp4. Hence, the depth at which dehydration reactions are expected to occur varies by almost 140 km within our parameter space. When comparing the reference model case2c_PvK and case2c_bc with constant thermal properties to the most complex model case2c_all that includes temperature-dependence of all thermal properties, the differences in depth are smaller ($\sim 28.8$ km). However, this is still a significant difference in predicted seismicity depth that could be important when comparing to data. Therefore, the predicted depth of intermediate-depth seismicity in thermal models of subduction should be viewed in light of the assumptions on the thermal parameters.

Additionally, previous thermal models (e.g., Syracuse et al., 2010; van Keken et al., 2012) that use constant values for the thermal parameters and reproduce a thermal structure that fits observed seismicity are expected to change when temperature-dependent thermal parameters are used, with implications for the thermo-chemical changes then ascribed to control intermediate depth seismicity. Depending on the choices of the functions describing the thermal parameters and their interaction, the fit with observed seismicity can change. To accurately determine the depth of intermediate-depth seismicity and the relationship between the thermal structure of the slab and intermediate-depth seismicity, we recommend the use of temperature-dependent

thermal parameters constrained by the insights on rock behaviour. Neglecting temperature-dependent thermal parameters could result in errors of tens of kilometres in the estimated depth of intermediate-depth seismicity or a misinterpretation of the relation between the thermal structure of a slab and observed intermediate-depth seismicity. When only one of the thermal parameters is temperature-dependent (i.e., just the thermal conductivity, or just the heat capacity) instead of all three, the differences in temperature profile are even larger, as their separate effects cancel each other out to an extent when considered simultaneously. Hence, to accurately model the thermal structure of subduction zone, especially at intermediate depth, the temperature-dependence of all three thermal parameters should be taken into account simultaneously.

### 4.1.2 Deep seismicity

The cause of deep earthquakes (>300 km) is hotly debated with proposed mechanisms such as dehydration embrittlement, transformational faulting, and (grain size assisted) thermal runway as a result of shear heating (see Green and Houston, 1995; Frohlich, 2006; Zhan, 2020, for an overview). Regardless of the exact mechanism responsible for deep earthquakes, it is clear that the thermal structure of the slab plays a large role through providing the optimal conditions for each of these mechanisms to occur in. In fact, recent studies by Jia et al. (2020); Liu et al. (2021) have shown that local slab temperature likely controls the rupture of deep earthquakes. Our results show that the effect of using temperature-dependent thermal parameters instead of constant values grows more pronounced with depth. Therefore, we expect that adding temperature-dependent thermal parameters will significantly affect models of the thermal structure of slabs at depths between 300–600 km relevant to deep earthquakes.

### 4.1.3 Megathrust seismicity

The spatial extent of megathrust seismicity and, in particular the maximum potential earthquake magnitude for interface events in a subduction zone, are determined by the size of the seismogenic zone (and hence the maximum rupture width) bounded by empirically-derived updip and downdip limits (e.g., Heuret et al., 2011). The updip limit of the seismogenic zone is usually thought to be determined by the transition from velocity strengthening in the shallowest part of the subduction zone to velocity weakening behaviour on the megathrust. However, it has been observed that large earthquakes such as the 2011 Tohoku-Oki earthquake can rupture through velocity strengthening materials, thereby increasing the seismogenic zone size Fujiwara et al. (2011); Lay et al. (2011). Similarly, tsunami earthquakes are thought to rupture the shallowest part of the subduction zone Kanamori (1972); Satake and Tanioka (1999); Lay et al. (2012); Satake (2015).

The downdip limit of the seismogenic zone is typically associated with the transition from brittle behaviour at lower temperatures to the onset of ductile behaviour at higher temperatures. The exact isotherms corresponding to this change in deformation style are still debated, with estimates ranging from $250°C$ to $550°C$ depending on the mineralogy Tichelaar and Ruff (1993); Peacock and Hyndman (1999); Scholz (2019). Most commonly though, the downdip limit of the seismogenic zone is associated with the $350°C$ and $450°C$ isotherms Hyndman and Wang (1993); Hyndman et al. (1997); Gutscher and Peacock (2003).

Using the constant slab dip of $45°$ in our model setup, we can then make a quick and rough calculation of the seismogenic zone width in our models if we assume that the upper limit of the seismogenic zone is the surface. For the reference model

case2c_PvK we observe depths of the 350°C and 450°C isotherms of 77.5 km and 110 km, respectively, which translates to a seismogenic zone size of 109.6 - 155.6 km. These rough estimates for the seismogenic zone size of our models are within the observed range (i.e., $\sim 75 - -250$ km ; Heuret et al., 2011). In the model where we take all temperature-dependent parameters into account, model case2c_all, the isotherm depths change by 3.8 km and 10.0 km for 350°C and 450°C, respectively, resulting in a maximum change in seismogenic zone size of 14.1 km.

However, our models here are of limited use in assessing the sensitivity of the temperature along the shallow subduction interface to the inclusion of temperature-dependent thermal properties for several reasons. First of all, in our simplified model geometry, the shallow dip of our interface is significantly larger than that typically seen in the interface seismogenic zone of most subduction zones (typically $23 \pm 8°$; e.g., Jarrard, 1986; Heuret et al., 2011; Schellart and Rawlinson, 2013). This translates to a typical depth of the downdip limit of the seismogenic zone of $40 \pm 5$ km Tichelaar and Ruff (1993). Besides that, we refrain from including the compositional complexity necessary to appropriately model the thermal structure of the overriding plate and a sedimentary forearc. Lastly, we do not include the effects of shear heating and fluid circulation on the shallow interface (England, 2018).

However, noting the impact that the variation in thermal properties at low temperatures (e.g., Figure 3) has on the rates at which cold material heats up near the top of the downgoing plate and in the wedge of the forearc, we recommend using temperature-dependent thermal parameters in thermal models of subduction zones, in addition to the other influences mentioned, for when physically realistic estimations of the seismogenic zone size are desired. Similarly, when observations are linked to the behaviour of the interface (e.g., limits on seismogenesis, on coupling, on slow slip, etc.), the inclusion of temperature-dependent thermal parameters may alter the inferred mineralogical and rheological controls on such transitions. Considering that we use an unrealistically steep subduction angle, our results on the changes in the maximum depth of the 350 and 450°C isotherms might underestimate the effect on the seismogenic zone size in realistic settings with lower megathrust angles.

## 4.2 Mineralogical evolution of the slab

As the subducting plate descends, it typically undergoes a range of mineralogical transitions, relating to the increase in pressure and temperature. These mineralogical changes, particularly the location at which dehydration reactions release fluids into the slab system, play a controlling role in determining the location of intraslab seismicity, and also in influencing a range of other geophysical phenomena, from the internal impedance and velocity contrasts within the slab (e.g., Abers, 2000; Rondenay et al., 2008), to the occurrence of slow slip events on the subduction interface (e.g., Peacock, 2009), to the development of a hydrated mantle forearc (e.g., Abers et al., 2017). The preservation of volatile-hosting lower-temperature material into the deeper mantle also plays a role in global geochemical cycles (e.g., Rüpke et al., 2004).

Whilst the kinematic constraints we impose on the slab in our models mean there is little variation in lithostatic pressure between models, we have shown that including the temperature dependence of thermal parameters in the modelling of slab thermal structures has an impact on the resultant temperature field. Whilst these changes are small relative to the total change in temperature experienced by the slab during subduction, they lead to a slightly different pressure-temperature evolution for

the slab material. An additional crustal layer in our models, for which we currently only employ a simple parameterisation, further alters the temperature field. We note that the changes in the temperature evolution of the uppermost $\sim 7$ km of the slab is particularly susceptible to the temperature dependence of thermal properties, given the rapid variation of such values at low temperatures (Figure 3). The mineralogical evolution of the shallowest part of the slab is therefore likely to be altered by the incorporation of temperature-dependent thermal properties, with initially more rapid heating at low pressures giving way to slower heating at higher pressures, in comparison to models using fixed, temperature-independent thermal properties. Dehydration reactions in hydrated basaltic oceanic crust typically take place between $350-450°C$, whilst those in serpentinised oceanic mantle concentrate between $600°C$ and $800°C$ (Hacker et al., 2003a). In linking geophysical observations to thermal models, we again note that the variation in depth of the $350°C$ and $450°C$ isotherms in our models of up to 38.8 km with respect to the reference model case2c_PvK (Figure 7) would translate for most subduction zones into a significant trench-perpendicular lateral shift. To a lesser extent, the same is true when merely considering the depth variation of $\sim 3.8-10$ km for the $350°C$ and $450°C$ isotherms between case2c_PvK and case2c_bc with constant thermal parameters and case2c_all with temperature-dependent thermal parameters, which represent our most simple and most complex models. This will have a significant impact on the source location of phenomena such as the migration of fluids from the slab to the forearc mantle and/or updip along the subduction interface.

Lastly, the model diagnostics we focus on here centre around the maximum depth of a given isotherm. However, the variation in thermal structure that we study will also impact on the thermal cross section of the slab at any given depth - with marginally colder slabs having a significantly greater volume of material below a given temperature at a given depth, and hence potentially altering the volatile fluxes within slabs into the mid mantle.

## 4.3  Model limitations and future work

With the exception of a different rheology in the mantle wedge, where we combine both diffusion and dislocation creep, we use the same model setup as the subduction zone community benchmark presented by van Keken et al. (2008). We choose this model setup, as it is well defined and documented and reproduced by many codes in the geodynamics community (see codes used in van Keken et al., 2008). Hence, we are able to study the effect of temperature-dependent thermal parameters on the thermal structure of subduction zones in an isolated, well-defined manner, although, as discussed, this does limit the direct applicably to observational data.

The model setup is greatly simplified and many complexities that are known to influence the thermal structure of the slab are ignored. As illustrated in the benchmark of van Keken et al. (2008) itself, one of the largest influences of thermal structure of the subducting slab is the employed rheology. The temperature model diagnostics in van Keken et al. (2008) change up to $189°C$ when changing from an isoviscous to a dislocation or diffusion creep rheology. To a lesser extent, the difference between a purely dislocation creep and diffusion creep rheology is noticeable with variations on the order of $10°C$ in model diagnostics. We find that employing a combined dislocation and diffusion creep rheology does not significantly change the model diagnostics compared to a purely dislocation or diffusion creep rheology. However, our approximation of combining a dislocation and diffusion creep rheology is simplistic. Using a composite rheology of diffusion and dislocation creep to

properly account for the nonlinearity of the two rheologies would be more physically appropriate (Ranalli, 1995; Karato, 2008; Gerya, 2019). This would likely introduce changes to the temperature field of the slab on the same order as the differences observed between a pure diffusion and a pure dislocation model as in van Keken et al. (2008). Similarly, it would be more appropriate to take into account the effects of the activation volume, which we currently set to zero. The activation volume is not well constrained (Karato and Wu, 1993; Dixon and Durham, 2018), but would affect the viscosity of the system and therefore indirectly change the flow patterns and thermal structure of the subduction zone.

Hence, the effect of using temperature-dependent thermal parameters in thermal models instead of constant values is a secondary effect to rheology when comparing isoviscous and non-linear rheologies (i.e., compare Figures S1-S4 to S5-S7). However, when comparing non-linear rheologies, using temperature-dependent thermal parameters instead of constant values will likely have a greater effect on the thermal structure of the slab than changing the details of the rheology formulation. Note that these conclusions relate to the thermal structure of the slab; the rheology plays a major role in the thermal structure of the mantle wedge and overriding plate, as evident from the original benchmarks presented in van Keken et al. (2008). Although our parameterisation of the overriding plate captures some of these complexities, our results cannot address all the complexities introduced by a rheology tailored to continental crust rocks in the overriding plate. Similarly, our overriding plate parameterisation does not capture the effect of crustal radiogenic heat production on the temperature field. In the absence of this, based on our results, we predict that changes in the model with regards to the overriding plate will not significantly affect the temperature field of the slab, although the overriding plate thickness would change the temperature depth distribution.

Apart from a simplified rheology, we also employ a simplified geometry in our model setup. Although the model serves as a good benchmark and we can infer some implications for seismicity from this simple setup, a strictly $45°$ dipping slab is not realistic. In nature the slab dip changes with depth with low dipping angles of $23 \pm 8°$ for the megathrust region (Heuret et al., 2011) and larger dip at depth (e.g., Isacks and Barazangi, 1977; King, 2001; Cruciani et al., 2005; Syracuse et al., 2010; Klemd et al., 2011; Hu and Gurnis, 2020). Therefore, more realistic models of the thermal structure of subduction zones include these complex geometries (e.g., Syracuse et al., 2010; van Keken et al., 2012). Our results indicate that in these complex models of the thermal structure of the slab, it is important to take the temperature-dependence of thermal parameters into account as well. Even though including them will likely not change the large-scale subduction evolution, it is important to include the temperature-dependent thermal parameters for accurate comparison with (earthquake) data.

Although we focus here on the effect of using temperature-dependent thermal parameters, there are numerous other processes relevant to the developing thermal structure of a subduction zone (see van Keken et al., 2019, for an overview). For example, frictional (or shear) heating along the plate interface (e.g., Peacock, 1992; England and Molnar, 1993; Peacock, 1993b; Peacock and Wang, 1999; Burg and Gerya, 2005; Gao and Wang, 2014, 2017) and radiogenic heating in the overriding plate (e.g., Gao and Wang, 2014; England, 2018) introduce additional heat sources to the system and result in warmer slabs in line with petrological estimates of the temperatures of rocks in a subduction zone (Penniston-Dorland et al., 2015). Typically these processes are included in models where a temperature-dependent density formulation is used, although the conductivity and heat capacity are often still taken to be constants. We deliberately do not include these additional heat sources when including the temperature-dependent density to isolate its effect on the thermal structure of a subduction zone. However, we recognise that

this may lead to thermodynamic inconsistencies, similar to those introduced through inconsistent thermodynamic potentials calculated from the thermal parameters (Schubert et al., 2001; Van Zelst et al., 2022). Phase changes, such as serpentinisation in the mantle wedge corner (e.g., Hyndman and Peacock, 2003) and the transition from blueschist to hydrous eclogite (e.g., Hacker et al., 2003a), also play an important role in establishing the thermal structure of the slab, as they are paired with the release of latent heat, density and subsequent volume changes, fluid production and heat advection (see Peacock, 2020, for an overview of petrologic models). Fluid flow and hydrothermal circulation within the upper part of the slab efficiently transport heat updip towards the trench (e.g., Spinelli and Wang, 2008; England and Katz, 2010; Faccenda et al., 2012; Rotman and Spinelli, 2013; Harris et al., 2017). Depending on the subduction velocity, this can significantly reduce the temperature of the subduction interface (Rotman and Spinelli, 2013). In line with this, processes such as melting and melt transport at the top of the slab and in the mantle wedge corner (e.g., England and Katz, 2010; Bouilhol et al., 2015; Perrin et al., 2016), magmatism (e.g., Jones et al., 2018), erosion (e.g., Royden, 1993; England, 2018), sedimentation (e.g., England, 2018), anisotropy (e.g., Morishige and Tasaka, 2021), and three-dimensional complexities (e.g., Gerya, 2011; Plunder et al., 2018; Wada, 2021; Qu et al., 2023) also play a role in the thermal structure of a subduction zone. In addition, subduction is an inherently time-dependent process with the temperature structure of the subducting slab likely changing throughout its evolution which is not captured by the steady-state thermal models presented here (King, 2001; Holt and Condit, 2021). Here, we deliberately choose to ignore these complexities to isolate the effect of temperature-dependent thermal parameters on the thermal structure of the slab. Future studies could focus on these processes to explore their effect on models of the thermal structure of the slab.

Although we consider a set of models where we include a simple parameterisation as an approximation of a crustal layer in the slab, our models are still restricted to a single composition. Hence, we do not explicitly include a crustal layer, and we neglect the impact of the mineralogical evolution of the slab on the temperature structure, both through the variation in thermal parameters with evolving mineralogy, and through the latent heat of mineralogical transformation. Our results suggest that including compositional heterogeneity, specifically oceanic crust, does not change the observed trends concerning the effect of temperature-dependent thermal parameters on the models. Hence, the main conclusions presented in this work are not affected by compositional heterogeneity. However, thermal parameters do vary greatly with composition (e.g., Whittington et al., 2009; Miao et al., 2014). Therefore, compositional variation plays a big role in controlling the thermal structure of the incoming plate (e.g., Richards et al., 2018), and therefore also the subduction zone dynamics (e.g., Gerya et al., 2004). Going forwards, the development of models that include both composition and temperature dependence in the thermal parameters is likely required to further progress the accurate modelling of the thermal structures of subduction zones.

Lastly, there are numerous functions describing the temperature- and pressure-dependence of the thermal parameters in the literature and existing functions are continuously updated with improved values for constants to better fit laboratory data. It is outside of the scope of this work to test all different formulations and here we follow McKenzie et al. (2005) and Richards et al. (2018) who used temperature-dependent thermal parameters for plate models of the cooling oceanic lithosphere. However, other possible functions of the temperature-dependence of thermal parameters include formulations from studies like Berman and Brown (e.g., 1985); Seipold (e.g., 1998); Hofmeister (e.g., 2007a); Wen et al. (e.g., 2015); Su et al. (e.g., 2018). In addition, pressure-dependent formulations have been proposed by Hofmeister (e.g., 2007b) and studies have shown that the residual

misfit between the model and the data reduces upon including the pressure-dependence of the thermal parameters (e.g., Grose and Afonso, 2013; Korenaga and Korenaga, 2016; Richards et al., 2018).

## 5   Conclusions

In this work, we look at the effect of using temperature-dependent thermal parameters in thermal models of subduction zones compared to using constant values.

We find that the use of temperature-dependent thermal parameters does not significantly affect the first-order thermal structure of a subducting slab, like the choice of rheology or plate age would. However, the local thermal structure of the slab does change on the order of tens of degrees when using temperature-dependent thermal parameters. This second-order effect could be significant for specific modelling applications that consider for instance seismicity and phase changes.

Using only a temperature-dependent conductivity decreases the temperature in the slab and results in a larger predicted

seismogenic zone width and deeper intermediate-depth seismicity with the maximum depth of the 600°C isotherm changing up to 87.5 km for a model using the thermal conductivity function of Hofmeister (1999); McKenzie et al. (2005) compared to a reference models with constant values.

Solely employing a temperature-dependent heat capacity has the opposite effect and results in a warmer slab with a shallower downdip limit of the seismogenic zone and predicted depth of dehydration reactions responsible for intermediate-depth

seismicity.

Using only a temperature-dependent density has the least effect on the thermal structure of the slab when compared to the reference model with constant values, although the slab is overall colder.

Combining the temperature-dependence of the three thermal parameters negates the effect on the thermal structure of the slab slightly, but the strong cooling of the slab produced by both the temperature-dependent thermal conductivity and density dom-

inates. Therefore, the modelled slab is colder than a slab modelled with constant thermal parameters with, e.g., the maximum depth of the 600°C isotherm changing by 28.8 km. In addition, the predicted surface heat flux increases when temperature-dependent thermal parameters are included. The importance of including temperature-dependent thermal parameters increases for increasing slab age, as the functions of the thermal parameters used in this paper have their most extreme values for lower temperatures.

Models including a parameterisation of oceanic crust in the slab and an oceanic or overriding plate show the same trends. We find that choosing consistent temperature boundary conditions is crucial and can otherwise lead to large differences in the temperature field of the slab. In contrast, the nature, or existence, of an overriding plate does not significantly affect the temperature field of the slab.

We caution that the conclusions drawn here stem from a highly simplified model of a subduction zone that is not suitable

for direct comparisons to nature and does not aim to reproduce observations. Hence, the details of our findings will likely change for more complex and realistic model setups. However, even considering the simplifications in our model setup, our results indicate that the changes in the modelled thermal structure of the slab, as well as the predicted surface heat flux will

have important implications on the estimated size of the seismogenic zone in these kinds of thermal models and on predictions where intermediate-depth seismicity might occur. These implications of the inclusion of temperature-dependent thermal parameters are of a lower order than the established first-order controls on the subduction zone thermal structure, such as the employed rheology and plate age. Nevertheless, we suggest that for optimal comparison to data and to avoid misinterpretations, temperature-dependent thermal parameters are also an important modelling ingredient that should be taken into account when using thermal(-mechanical) models of subduction zones, especially in studies that have seismological applications.

*Data availability.* The models were run with the open source code xFieldstone (Van Zelst, 2023, https://github.com/irisvanzelst/xFieldstone). The data used to reproduce the van Keken et al. (2008) benchmark and the Matlab scripts used for the postprocessing of the results and generation of the figures are shared in Van Zelst et al. (2023).

*Author contributions.* IvZ and TJC conceived the study. IvZ designed and ran the models, analysed the results, and wrote the article. CT and IvZ wrote the code xFieldstone. TJC supervised IvZ and contributed to the analysis of the models. All authors discussed the results and contributed to the model design and the final manuscript. The author order of the second and third authors was decided by a coin flip.

*Competing interests.* The authors have no competing interests.

*Acknowledgements.* We thank editor Taras Gerya and six anonymous reviewers for constructive feedback that helped to improve this manuscript.

We thank Peter van Keken for providing the original data from van Keken et al. (2008) to benchmark the code presented here and for providing additional insights on the model setup.

Elements of this work were undertaken on ARC4, part of the High Performance Computing facilities at the University of Leeds, UK. We also ran simulations on the Plejades work stations of the German Aerospace Center (DLR), Germany. IvZ and TJC were funded by the Royal Society (UK) through University Research Fellowship URF\R1\180088 and Research Fellows Enhancement Award RGF\EA\181084. TJC was also supported through COMET, the UK Natural Environment Research Council's Centre for the Observation and Modelling of Earthquakes, Volcanoes, and Tectonics. IvZ additionally acknowledges the financial support and endorsement from the DLR Management Board Young Research Group Leader Program and the Executive Board Member for Space Research and Technology.

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
