# Peer review of "The effect of temperature-dependent material properties on simple thermal models of subduction zones"

_EGUsphere, 2022_

## Author Comment (AC1)

Dear Editor, dear Reviewers,

Please find enclosed our revised manuscript (Preprint egusphere-2022-768) on 'The effect of temperature-dependent material properties on simple thermal models of subduction zones' by Van Zelst et al.. We thank the two reviewers and the editor for their detailed feedback. We incorporated most of their suggestions in the new manuscript, which we believe has resulted in a clearer and more nuanced paper.

The two reviewers had different views on the direction this manuscript should take to ready it for publication. Reviewer 1 requested that we use a more complex, realistic subduction zone geometry in our model setup. In contrast, reviewer 2 suggested that we could cast this paper and the current results as a companion paper to Van Keken et al. (2008) and/or present our work as a negative test. We have decided to largely follow the suggestions by reviewer 2 and emphasize our closeness to the original Van Keken et al. (2008) paper. Although we cannot fully cast our work as a negative test, because our models show that temperature-dependent thermal parameters do have an effect on the resulting thermal structure of the subducting slab, we now better explain for which purposes these second-order effects are likely relevant. We also explicitly mention parameters with first-order effects on the thermal structure of a subduction zone, such as rheology or plate age, to avoid any confusion as to the relative significance of our findings.

By following the suggestion from reviewer 2, we cannot simultaneously follow the suggestion of reviewer 1 of a more complex, realistic subduction zone geometry. This is unfortunately beyond the scope of this paper, which we explain in more detail in the rebuttal letter below. However, we have addressed their other comments and thereby believe we have improved the paper sufficiently for consideration for publication.

We also note that reviewer 2 commented that the paper was quite lengthy and should perhaps be shortened and more limited in scope. We refrained from doing this, as a lot of the added material that increased the length of the initial manuscript stems from previous rounds of revisions at JGR: Solid Earth. Since we also want to honour the valuable input from those two reviewers, we have not shortened or limited the scope of our paper. We believe that the extensive discussion is useful to provide the context in which our results are meaningful, both in terms of discipline (i.e., a seismological application) and our assumptions and model limitations.

Responses to the suggestions by the reviewers are indicated in green. Line numbers refer to the line numbers of the provided tracked changes file.

Thank you for considering this revised manuscript for publication.

Yours sincerely,
Iris van Zelst (corresponding author),
Cedric Thieulot, Timothy J. Craig

**Reviewer 1**

I reviewed an earlier version of this manuscript that was submitted to another journal. I was unenthusiastic recommending publication because 1) the presentation in parts was very sloppy; 2) claims were made that the effect of including T-dependent properties was large whereas it was demonstrated in the paper that the effects were secondary compared to other governing parameters such as plate age or convergence velocity; and 3) that the (benchmark) model geometry and description used was unsuited to make inferences about thermal structure of subduction zones (even if it might be a useful geometry to test geodynamical codes).

The presentation has improved (but not completely, see below) and some of the most dramatic statements in the previous manuscript that suggested great importance of the T-dependence of the parameters in the heat equation have been removed, at least from the first parts of the paper. There are still quite a few (albeit repetitive) statements that I think are a mischaracterization of your findings (see below). You demonstrate that the thermal effects that you study are anything but secondary, if not tertiary, even when looking at the possible location of the BDT, compared to variations in the main driving parameters (slab age, speed, and dip). I will expand on my remaining concerns below.

As such I cannot recommend publication in present form. I realize a lot of work (and computer time and CO2 production) has gone into this paper. I could possibly be convinced that a revised version could be acceptable if a) the authors would phrase their modeling as a negative test of the hypothesis (because they demonstrate that T-dependence of k, c_p, and rho are minimal compared to the reference case of constant parameters; see below); and b) either a more realistic subduction geometry were to be used (see below) or that the heat equation would be solved as a time-dependent one with an evolution to 40 Myr or so – that should be enough to mitigate the pronounced negative effects of the benchmark model assumptions. As for b) I would prefer the former as then you can also include (more) realistic radiogenic heating and a more realistic wedge boundary condition for temperature.

I'll provide more details on my main two criticisms of this paper followed by a chronological list of issues that I think require attention below.

It is clearly demonstrated in the figures that the importance of T-dependent k, c_p, and rho, their effects are secondary at best. The cause for this is shown in Figure 2: the variations in the thermal range of interest (i.e., 400 C and above) are limited to 10-20%. The largest differences are near 0 C but this is not a temperature of great interest to subduction zone thermal modeling (except perhaps in the top boundary condition). The effect on the thermal structure of the incoming lithosphere is modest – the maximum difference at any given depth is a little hard to guess because of the graphics but it looks like 30 C or so. Rather minor compared to what you get when you change the age of the incoming lithosphere.

While there are some cases in Figure 7 that, side by side, suggest relatively large shifts in the depth of contours (e.g., 'case2c_k1' vs. 'case2c_cp3') there appears to be a minimal shift between the reference model ('case2c_PvK') and the model incorporating the T-dependence in all parameters of interest ('case2c_all'). The same is illustrated in Figure 9, where the maximum

change is perhaps 40 km. That is minimal compared to the shift in isocontour depth that occurs when changing the slab age (as is shown nicely in this Figure). Clearly, the T-dependent variations in k, c_p, and rho are secondary (if not tertiary) to other subduction zone parameters such as slab age (shown here) and convergence speed (easily predicted by way of the thermal parameter).

We agree with the reviewer that the effect of temperature-dependent thermal parameters is not a first-order effect on the thermal structure of subduction zones and indeed takes a backseat compared to the choice of rheology or plate age. Nevertheless, the changes incurred by using temperature-dependent thermal parameters are significant in certain applications, such as comparison with earthquake hypocenters or when considering the exact depth of phase changes. Other applications, focussing for example more on the large-scale subduction dynamics of a certain region would indeed likely find that the effect of temperature-dependent thermal parameters is irrelevant. We have rewritten parts of the manuscript (see below) to make this distinction more clear and indeed emphasise that temperature-dependent thermal parameters are of less importance than first-order modelling ingredients such as rheology and plate age.

I do not understand why the authors use this 'highly simplified' geometry with 'simplicity' (L135). I would say the model geometry and parameter assumptions are overly simplified and very far away from a 'generic' subduction zone (L140). There is no subduction zone on Earth that dips under a 45 degree angle to 600 km depth or that has no radiogenic heating in the overriding crust. Most geophysical observations exclude coupling at 50 km depth (e.g., Wada and Wang, Gcubed, 2009). The model geometry may be useful for benchmarking, but there is a huge artefact that occurs with temperature-dependent viscosity which is the formation of a very large and unrealistic 'viscous belly' (e.g., Figure 4c). This is a consequence of the assumption of steady-state which causes progressive cooling of the overriding lithosphere that effectively takes place over hundreds of millions of years and its thickness is enhanced by the lack of radiogenic heating in the overriding crust (see discussion in Hall, PEPI, 2012). Most subduction zones don't exist for that long and heat flow observations or observations of seismic attenuation clearly show that such a viscous belly does not exist (where we have such observations).

We would like to emphasise that we do not at any point claim that we are modelling a generic subduction zone, because our model setup can indeed not be compared to any realistic subduction zone setting, as noted by the reviewer. Rather, we employ the 'generic modelling philosophy', as defined in Van Zelst et al., (2022), where we aim to better understand the subduction zone system's general behaviour and physics (i.e., rather than aiming to reproduce the specific state of any one subduction zone on Earth). We believe that throughout the paper we point out to the reader sufficiently that our model setup is highly simplified and the results and conclusions should be considered in light of our simplifications and assumptions (see, for instance, our title, abstract, and lengthy discussion on model limitations).

As such the variations in various figures in the lithosphere look much larger (see e.g., Figure 6a) than they will be in any (more) realistic subduction zone geometry. I predict that the temperature

variations in the overriding plate will be restricted to the shallowest and coldest portions of the crust if more realistic subduction zone model parameters (as in, e.g., Wada and Wang, 2009; other papers cited in the ms.) were used. I do not know what the consequences for the thermal distribution in the slab will be, but they won't be completely insignificant. I think it is essential that the authors demonstrate that their conclusions still stand with a more realistic set of assumptions of the base model (including geometry, coupling point, wedge viscosity, radiogenic heating in the overriding crust, etc.).

This is unfortunately outside of the scope of this manuscript. We trust that our extensive discussion as well as the mentions throughout the manuscript and abstract clarify under which assumptions our conclusions are valid.

**Comments**

L51ff. Many of the papers cited do not study the 'thermal evolution of a subduction zone in steady state'. Many of these use time-dependent modeling. Please fix.

We have removed "in steady state", for clarity.

L54, 60, other places. Please pick an upper case or lower case for references to 'van Zelst', 'van Dinther' or 'Van Keken' and stick to it.

The correct capitalisation of Dutch surnames is unfortunately something that (English) journals do not take into account, as the capitalisation of the word 'van' depends on whether or not there is something (a first name or initial) in front of it. Most journals therefore use the (incorrect) spelling of, e.g., 'van Dinther' in a citation, where it should be 'Van Dinther'. I have now chosen to use the (incorrect) non-capitalised versions for Van Dinther and Van Keken citations, because that aligns with the journal's preferred referencing format and I don't want to make assumptions on how they would prefer to have their name spelled. However, for myself, I use Van Zelst (i.e., with a capital V), as I find it important that my name is spelled correctly.

Eq. (2). Why include the gravity term if you set it to zero? I don't think this equation follows the benchmark paper because of this reason.

We initially included the gravity term for completeness, but since it is equal to zero we have now removed it from the equation. We solve the same equations as the original benchmark paper by Van Keken et al (2008).

L159. This seems like a large waste of computational resources. Why not solve the Stokes equation in the mantle wedge. Your code appears to be highly inefficient (you should be able to solve the benchmark cases in minutes on a single core of a laptop using existing codes but you appear to need to use Arc4 and German supercomputing resources) and making it even more inefficient by first solving the Stokes equation and then overwriting it with a kinematic condition just doesn't make any sense (at least, not to me).

The code is indeed inefficient as it has been adapted from the educational code FieldStone. The way the code is written is therefore easily readable and understandable to non-experts, although this comes at the cost of efficiency. Since we will share the code upon publication, we value high readability in our code. In terms of computational resources, the code could also have been run on a simple desktop or laptop, but due to the pandemic I did not (and still do not) have access to a work desktop or laptop. Therefore, we used the available local clusters of the University of Leeds and DLR.

L159. Do you really solve the Stokes equations in the crust? I am amazed you are getting a decent comparison to the actual benchmark, which imposes a zero slip boundary condition at 50 km depth (away from the slab).

We indeed solve the Stokes equation in the crust, as we solve the Stokes equation in the entire domain. However, we also impose a no slip boundary condition at the bottom of the overriding plate (line 154-155). Together with the other boundary conditions, this results in the solution of the Stokes equation to be automatically 0 in the overriding plate and us recovering the solution of the benchmark. We follow the same procedure as in van Keken et al (2008).

L286, other places. Why do you use -half- the value of the computed conductivity? That seems excessive. What paper suggests that this is reasonable? The conductivity in the crust should be lower but typical values are generally only 20% lower than that of the mantle.

A value of half that of the mantle is appropriate at lower temperature conditions (suitable in the crustal layer, based on both mineral physics calculations (e.g., Grose and Afonso, 2013), and ocean core sample observations (e.g., Kelemen et al., 2004), which typically find values of 2 – 2.5 W/m/K, in comparison with values for olivine at such temperature of 4 – 5 W/m/K). We have added these references in the text. We also note that some convergence at higher temperatures is likely, but during the initial stages of subduction, a 50% reduction from the values for a pure-olivine mantle seems a reasonable approximation for an oceanic crustal aggregate.

L343. I do not find it surprising at all that you get 'distinct differences' (even if they are 'outside the main focus [region?] of your study') because you use a different wedge rheology. This whole paragraph seems unnecessary.

Indeed, it is not surprising that we get distinct differences due to the different rheology. However, we believe it is important to highlight these differences nonetheless, in case anyone wonders why they are there.

L395. "the extreme effect in the overriding plate" Maybe I'm misunderstanding you here but I can't see how a temperature difference of 20 C (Figure 6a, others) is 'extreme'.

We reformulated this.

L412. I totally agree that the results are 'unrealistic' because of the 'artificial boundary effects'. That should give it away that this is not a generic subduction zone but much simplified model set up to allow for a simple benchmark comparison. This model should not be used for any other research purposes.

As mentioned above, we never mention that we are modelling a generic subduction zone. Instead, we say we model a highly simplified and unrealistic subduction zone setup that follows the generic modelling philosophy of numerical modelling (Van Zelst et al., 2022). These models can be very insightful when determining the general, physical behaviour of a system and are therefore well-suited for research purposes.

First paragraph of discussion: I cannot see how you can call 20 C or a change in depth of a contour by a few 10s of kilometers 'significant' or 'great'. There is a change, yes, but it is secondary compared to changes in more important driving factors of subduction zone thermal structure.

We have rephrased this paragraph (line 464-471). The reviewer is correct that there are more important driving factors of subduction zone thermal structure; we hope we make this clear in the manuscript now. However, the changes induced by temperature-dependent thermal parameters are still large enough to be important when aiming for as-accurate-as-possible thermal models of subduction zones when comparing, for instance, to observed seismicity.

L502. "Neglecting temperature-dependent thermal parameters could result in significant errors of up to hundreds of kilometers in the estimated depth …." You really do not show this anywhere in the paper. A person reading just the abstract and this part of the discussion (because perhaps this person is only interested in seismicity) would walk away with a thoroughly misled impression of your paper.

We have reformulated this (line 516-517).

L512ff. More of the same. Just stating that things are significant doesn't make them change from being (relatively) insignificant. You have not demonstrated this at all. Sorry to be repetitive, but I find it is necessary to call out repetitive mischaracterizations of your own work essential.

See our response above; the changes are not first-order but still significant enough that they should be taken into account.

L572ff. This is not an original finding is it? I believe it is even in the benchmark paper.
This is correct and this is why we refer to the van Keken et al (2008) benchmark paper when making this statement.

L590. You seem to be repeating statements from an earlier paragraph. Irrespective, I wholeheartedly agree that you should not be using a subduction geometry that has a continuous dip of 45 degrees.

In this paragraph, we put our assumption into context so readers know how (un)realistic our assumption is, such that they can view our results and conclusions through the correct lens. As mentioned before, a different model setup is unfortunately outside of the scope of this study.

L649. This is completely cherry-picked. You choose a complete outlier that is based on a very selective comparison of extreme end-members of models. You clearly show that the variations between the reference case and your preferred case are minimal.

This is not cherry-picked at all. Our preferred case is indeed the model where all thermal parameters are temperature-dependent and we present our conclusions on this model in a later paragraph (line 684-690). Indeed, the preferred model is also the model we mention in the abstract. However, for the completeness of the conclusions, we briefly discuss the individual effect of having temperature-dependent thermal conductivity, heat capacity, and density. When we include a temperature-dependent thermal conductivity, the effect on the 600 C isotherm is as big as stated here. In the rest of the conclusions, we also present the effect of the temperature-dependent heat capacity, and density, as well as our preferred model and other model batches that we ran (i.e., the effect of slab age and the effect of a crustal parameterisation). We therefore do not cherry-pick at all, but instead clearly state all our findings in a succinct summary. To clarify this in the text, we have slightly rephrased the conclusion.

L670ff. I totally agree. I think you should explore this.

This is unfortunately outside the scope of this study.

L673ff. Aside from the slightly awkward styling of the sentence, you do -not- show that temperature-dependent thermal parameters are an important modelling ingredient. See Figure 9.

We rephrased this sentence. However, we still believe that - based on Figure 7 and 9, for instance - we show that temperature-dependent thermal parameters are an important modelling ingredient. We clarify in the manuscript that temperature-dependent thermal parameters do not affect the thermal structure of subduction zones to first order, but the changes of a few tens of degrees that we observe and therefore a few tens of kilometers of the expected isotherm depth are indeed important, especially for seismological applications or interpretations of these models.

Data availability statement. I do not know why one can submit a paper without making the "data" (or in this case models) available. Making them available after publication doesn't allow for an evaluation of said "data" or models, at least not until after the fact.

We are happy to make the data available to reviewers in advance of publication by sending a .zip file or something similar. Unfortunately, we cannot upload this to the Solid Earth

environment and Zenodo doesn't allow changes after publication, so in order to avoid uploading it twice, we made the data availability statement that we did.

References. I appreciate you cleaned up some of the most egregious mistakes in the previous ms. that I saw but a bunch of remaining ones are easily spotted particularly in capitalization, lack of correct typography, and spurious / missing information ('Geophysical research letters'; 'https://doi.org/xxx'; 'H2O'; L886-887), article numbers that are confused with page numbers (L774, L799, others), or incomplete (L790) and nearly completely incomplete citations (L740).

We went through the bibliography again and cleaned it up.

I'm surprised that the authors do not seem to be aware of Chemia, Dolejs, and Steinle-Neumann, JGR, 2015. Seems like a highly relevant reference here.

We added this reference.

**References**

Grose, C. J., & Afonso, J. C. (2013). Comprehensive plate models for the thermal evolution of oceanic lithosphere. Geochemistry, Geophysics, Geosystems, 14(9), 3751-3778.

Kelemen, P. B., Kikawa, E., Miller, D. J., Abe, N., Bach, W., Carlson, R. L., ... & Takazawa, E. (2004). Proceedings of the Ocean Drilling Program, Initial Reports, Volume 209, Drilling mantle peridotite along the Mid-Atlantic Ridge from 14 degrees to 16 degrees N, Sites 1269-1275.

Richards, F. D., Hoggard, M. J., Cowton, L. R., & White, N. J. (2018). Reassessing the thermal structure of oceanic lithosphere with revised global inventories of basement depths and heat flow measurements. Journal of Geophysical Research: Solid Earth, 123(10), 9136-9161.

van Keken, P. E., Currie, C., King, S. D., Behn, M. D., Cagnioncle, A., He, J., ... & Wang, K. (2008). A community benchmark for subduction zone modeling. Physics of the Earth and Planetary Interiors, 171(1-4), 187-197.

Van Zelst, I., Crameri, F., Pusok, A. E., Glerum, A., Dannberg, J., & Thieulot, C. (2022). 101 geodynamic modelling: how to design, interpret, and communicate numerical studies of the solid Earth. Solid Earth, 13(3), 583-637.

---

## Author Comment (AC2)

Dear Editor, dear Reviewers,

Please find enclosed our revised manuscript (Preprint egusphere-2022-768) on 'The effect of temperature-dependent material properties on simple thermal models of subduction zones' by Van Zelst et al.. We thank the two reviewers and the editor for their detailed feedback. We incorporated most of their suggestions in the new manuscript, which we believe has resulted in a clearer and more nuanced paper.

The two reviewers had different views on the direction this manuscript should take to ready it for publication. Reviewer 1 requested that we use a more complex, realistic subduction zone geometry in our model setup. In contrast, reviewer 2 suggested that we could cast this paper and the current results as a companion paper to Van Keken et al. (2008) and/or present our work as a negative test. We have decided to largely follow the suggestions by reviewer 2 and emphasize our closeness to the original Van Keken et al. (2008) paper. Although we cannot fully cast our work as a negative test, because our models show that temperature-dependent thermal parameters do have an effect on the resulting thermal structure of the subducting slab, we now better explain for which purposes these second-order effects are likely relevant. We also explicitly mention parameters with first-order effects on the thermal structure of a subduction zone, such as rheology or plate age, to avoid any confusion as to the relative significance of our findings.

By following the suggestion from reviewer 2, we cannot simultaneously follow the suggestion of reviewer 1 of a more complex, realistic subduction zone geometry. This is unfortunately beyond the scope of this paper, which we explain in more detail in the rebuttal letter below. However, we have addressed their other comments and thereby believe we have improved the paper sufficiently for consideration for publication.

We also note that reviewer 2 commented that the paper was quite lengthy and should perhaps be shortened and more limited in scope. We refrained from doing this, as a lot of the added material that increased the length of the initial manuscript stems from previous rounds of revisions at JGR: Solid Earth. Since we also want to honour the valuable input from those two reviewers, we have not shortened or limited the scope of our paper. We believe that the extensive discussion is useful to provide the context in which our results are meaningful, both in terms of discipline (i.e., a seismological application) and our assumptions and model limitations.

Responses to the suggestions by the reviewers are indicated in green. Line numbers refer to the line numbers of the provided tracked changes file.

Thank you for considering this revised manuscript for publication.

Yours sincerely,
Iris van Zelst (corresponding author),
Cedric Thieulot, Timothy J. Craig

**Reviewer 2**

The objective of this paper is to understand the variability of the thermal structure along the subduction interface when using a specific heat capacity, conductivity and density are temperature dependent. The temperature dependence in these parameters has not been considered in previous (steady-state) subduction zone simulations (to my knowledge). The authors conduct their analysis using a model inspired by the reference model defined in a community benchmark paper (van Keken et al. (2008)).

I do not consider this paper appropriate to publish in its current form for the reason that the conclusions and numerous statements made in the paper are not supported by the results shown. Worse over, the authors actually appear to contradict their own findings throughout the paper on several occasions.

We have reformulated sentences to ensure that there is no ambiguity and perceived contradictions in our writings. By clarifying our writings, we now also ensure that everything is clearly supported by the results shown. See below for specific changes made in response to comments by the reviewer.

The closing sentence is one example: "For optimal comparison to data and to avoid misinterpretations, we therefore suggest that temperature-dependent thermal parameters are an important modelling ingredient and that they should be taken into account when using thermal(-mechanical) models of subduction zones."

- Your own results actually show the assumption of steady-state (+ age) has a much larger influence on the temperature than including temperature dependence in the thermal coefficients (\rho, C_p, k).

This is correct. However, we hope to convey that temperature-dependent thermal parameters are also an important modelling ingredient to take into account when one wants to accurately model the thermal structure of a subduction zone. Especially for applications where modellers compare with observed seismicity, changes in the temperature field of tens of degrees or tens of kilometers are significant. To clarify that our findings are not first-order controls on the thermal structure of subduction zones, we have added a sentence in the conclusions (line 700 - 702).

- You neglect shear heating. Including that shear heating alone has been reported to increase the temperature by > 200 deg C, see for example Peacock, Geol. Soc. Am. Bull., (1993); England and Molnar, Tectonics, (1993); Burg and Gerya, Geology, (2005).Your results appear to indicate that the temperature dependence results in +/- 20 deg C variations in the thermal structure along the subduction interface. Given that the two points above, the temperature dependence you've introduced seems to be rather a secondary effect and thus the claim that T-dependence should be taken into account for reasons of accuracy, realism and to avoid misintrpretations in data / obsertvations is unjustified and unsupported. As written in its current form I found this contribution unclear, ambiguous and often disingenuous.

As mentioned above, we would like to convey that temperature-dependent thermal parameters are also an important ingredient to take into account when modelling the thermal structure of subduction zones. We do not claim that they are the most important ingredient - indeed, as the reviewer mentions, there are many other more important first-order factors, including the rheology, plate age, and the addition of important thermal processes such as shear heating. However, we show that on a smaller scale, temperature-dependent thermal parameters are indeed something to take into account, especially when one wants to use the models for comparisons with seismicity: then these second-order effects to thermal structure become important.

To make it more clear that we are talking about a second-order effect on thermal structure throughout the manuscript, we have reformulated the text throughout the manuscript (also see comments below).

We already mention the effect of shear heating on the thermal structure of subduction zones and the fact that we neglect it in our model in line 624. We have added the additional references that the reviewer mentions.

With our extensive discussion on model limitations and other model ingredients that could affect the thermal structure of a subduction zone, we believe we convey to the reader the nuance and caveats of our results concerning the important, though secondary, effect of temperature-dependent thermal parameters in thermal models of subduction zones.

From your results, the inclusion of the T-dependence does not appear to greatly influce the temperature along the interface. Hence, instead of exaggerating or extrapolating the results you have, it would be better to just report / document what you find. That is, I suggest you refactor the submission such that it is more like a companion paper to van Keken (2008) which only quantifies the thermal variability — within the scope of the idealised subduction model you consider — due to introducing temperature dependence. Focus on reporting the facts which are supported by your results, and place them within the context of all other modelling assumptions which are made in your idealised subduction model. In my opinion, confining the scope of the study in this way would make it a better contribution.

With the changes to the text we made based on the recommendations of the reviewer, we now believe that we are unambiguously reporting the findings of our study. We indeed view our paper as a companion paper to Van Keken et al (2008) to an extent, but we believe that the extensive discussion with potential implications and model limitations is beneficial for readers who would like a more in-depth view of the applicability of our results. In addition, many parts of the paper were added on the request of previous reviewers at JGR: Solid Earth and we are reluctant to remove those additions, since they also provided valuable comments that improved the paper. Hence, we have mostly focused on reformulating parts of the discussion and our conclusions to align with the current reviewer's wish of confining the scope of the study.

**Comments**

L77: "well-constrained" - In what sense is the community benchmark model well-constrained? I agree its simplified and well-defined. But it's not well-constrained.

We changed it to well-defined (also at other instances in the text).

Eq (2) Why bother to introduce \vec g and then promptly state its value will always be zero?

We removed the second term in this equation and now state that we solve the conservation of mass and momentum with the assumption of zero gravitational acceleration, i.e., without introducing **g** in the text.

L111 The statement "purely viscous rheology and hence neglect any elastic and plastic contributions to the deformation." seems redundant. You can just say you consider a "purely viscous rheology".

We changed this.

L112 You relate the deviatoric stress to the deviatoric strain-rate. You aren't relating the "stress to deformation" at all.

We reformulated this.

L121 "assume zero activation volume". The importance of making this assumption, i.e. what effect / influence this has on the thermal structure is not at all cosidered or discussed. That seems like an oversight. Just beacuse the assuumption was made in the benchmark paper doesn't mean it's an appropriate choice for subduction modelling in general.

We reformulated our initial statement in the methods where we said 'assume zero activation volume' to clarify why we make this assumption. In addition, we have added a few sentences to the discussion to clarify what the effect of a non-zero activation volume would be on our model results (line 598 - 601).

L127 You defined something as "square root of the dev strain-rate tensor" then re-defined it is the "effective deviatoric strain rate". Just provide one definition and remove "i.e., effective deviatoric strain rate".

We removed the explanation between brackets on the effective deviatoric strain rate.

L137 In what sense is the benchmark well-constrained? Such a term would be interpreted to mean that the definition of the model is somehow in agreement with a natural subduction zone (which it is not).

We changed it to well-defined (see above).

L164 "temperature compared to the previous iteration change less than a given tolerance " this stopping condition will will return a false positive if no progress is made in solving the non-linear problem.

This is true, but we observe monotonic convergence for the velocity (vx and vy) and temperature in the models. Therefore, the scenario where there is a false positive of convergence due to no progress in the solving of the system, is not one that occurs with our model setup.

As an example of what we observe concerning the convergence of the model, here is the convergence plot of the model case2c_all:

[Figure]

The chosen tolerance (1e-5) in essence means that we solve the system until the average value of the temperature field change between two consecutive iterations does not change more than 0.01K, which is sufficient for this type of model setup. As mentioned in the paper, additional tests show that employing a lower tolerance of 1e-3 (which is always reached before 50

iterations) changes the model diagnostics from the results section by less than 1°C and has no effect on the reported isotherm depths.

L171 This statement "results in robust convergence" is completely false and should be removed. Your stopping condition (as mentioned above) doesn't monitor the convergence of the solution to the nonlinear problem F(v, T) = 0. Hence you cannot infer convergence is "robust". Using your stopping condition, when the non-linear solver residuals stagnate, (meaning no progress is made) you will incorrectly interpret this as converged.

We changed it to "which prevents numerical oscillations in the solution towards convergence".

Eqns 13-16 define the solution procedure for a linear problem (i.e. when \rho, C_p and k are not functions of T). You stated earlier you incorporate the nonlinear parameters into this 1D model and use them as boundary conditions. Please correct the description of the method used to obtain the 1D temperature profile for the non-linear case.

The current description accurately describes what we do in the code. For this procedure we followed Richards et al. (2018). As the reviewer points out, it is true that we do not perform any additional non-linear iterations beyond the described predictor-corrector step, similar to Richards et al. (2018). We also note that McKenzie et al. (2005) used time steps on the order of ~7000 years and only performed 2-4 non-linear iterations.
Following the reviewer's question we investigated the matter by looking in detail at the influence of the CFL condition to see if we make any errors by neglecting additional non-linear iterations. We find that due to our low time step (delta t = 1000 years), we are well below the CFL condition. We find that our choice of low time step catches the changes that non-linear iterations would have provided. To illustrate the small changes induced by the use of different time step values, and hence by proxy the addition of non-linear iterations, we show the temperature profile (with depth and zoomed in) of the case2c_all model at 50 Myrs calculated with different time steps: dT = 500 years (red + dotted), dT = 1000 years (green + dashed), and dT = 2000 years (blue + solid). The changes in temperature for a given depth are on the order of 0.005 C and therefore negligible.
Hence, we keep the description of our method as is, as it accurately reflects what we did. Due to the small time step, we did not need to perform any additional non-linear iterations, which is why they are not mentioned in the text.

[Figure]

[Figure]

Eq (20) Don't use \cdot to denote multiplication. You have used the same notation to denote a dot product (e.g. Eq (1)).cc

Since we use \cdot in many of the equations to denote multiplication and make the equations more readable, we keep \cdot to denote multiplication. However, to distinguish it from the dot product in equations 1-3, we now use a thicker dot (\bullet) for those equations.

L275 Why is this statement even made? Your point was made clearly when you wrote down Eq (3) without using \kappa.

We removed this statement.

L315: The L_2 norm (big L2) defines an integral. l_2 (little L2) is used to define a sum. Please correct this.

Thanks for pointing this out. We corrected this.

Figure caption 4. You state the velocity contours go up to 5 m/s - I think you mean 5 cm/yr. Actually throughout this caption you speak about velocities measured in m/s which is incorrect. Same comment for figure captions 5 and S1.

You are indeed correct. We fixed it.

Figure 7: The plot style is inappropriate. When you connect dots together with a line you imply there is a relationship between the two data points. However, the x-axis in this plot are different models - hence there is no relationship between the data (e.g. between all the yellow squares for example). Remove the lines connecting the data points.

This is an excellent point. We have removed the connecting lines and slightly changed the colours of the figure to make the entire figure more readable. We have updated figures 7, 9, and S29 in the supplementary material, as they all shared this problem.

L394 Here you say "extreme effect in the overriding plate indirectly affects the thermal structure of the slab." whilst at L 452 you say "… the nature of the overriding plate, and indeed the inclusion of an overriding plate at all, does not significantly affect the temperature field." You supported this statement with figure 7.

The difference between these two statements is that in the first statement, we discuss that the temperature in the overriding plate is affected the most by the inclusion of temperature-dependent thermal parameters, which indirectly affects the thermal structure in the subducting slab. In the second statement, we talk about the effect of changing the nature of the overriding plate (continental plate, oceanic plate, etc) on the thermal structure of the subducting slab. Hence, there is no contradiction here, as we were talking about two different things. This

14

was worded confusingly, so we have rewritten both statements to make this distinction more clear (line 399 - 401 and 459 - 461).

L460-465 You state "Our models with different plate ages show that the implications generalise to ALL subduction zones regardless of plate age but still lack realism…" Your results do not support your implications generalise to ALL subduction zones - at best it generalises to those which have a constant dip of 45 degrees and are at steady state.

We reformulated this to "Our models with different plate ages show that our conclusions are valid regardless of the slab age"

L503-504 You write "Neglecting temperature-dependent thermal parameters could result in significant errors of up to hundreds of kilometres in the estimated" but none of the results presented in the paper support this statement. Take figure 7 and compare the two extreme models case2c_PvK_cp and case2c_all which you regard as the least inconsistent and the most self-consistent / accurate / complex. The 600 degC isotherm  is shift by ~25 deg C (squares) and ~ 50 deg C (circles). It's not changing by 100's. of deg C. The differences are even smaller for the 350 and 450 deg C isotherms.

We changed this sentence to "tens of kilometers" and we added some extra sentences to provide more nuance to this statement by discussing the difference between considering all 3 thermal parameters to be temperature-dependent and including only 1 thermal parameter that is temperature-dependent.

L588-589 Regarding "However, based on our results, we predict that changes in the model with regards to the overriding plate will not significantly affect the temperature field of the slab." - yes I agree, assuming the over-riding plate parameterization did not include any radiogenic heat production.

This is a good point. We added a sentence to explicitly mention our lack of radiogenic heat production in the overriding plate, so our prediction is now written with that caveat beforehand.

L645-650 Here you state you have 87.5 km variation (it looks more like 50), however for the cooler isotherms (where the variation is actually less!) you previously reported 100's of km of variation (L503-504).

We changed the previous statement; the value reported in the conclusions of 87.5 km is correct.

**References**

Grose, C. J., & Afonso, J. C. (2013). Comprehensive plate models for the thermal evolution of oceanic lithosphere. Geochemistry, Geophysics, Geosystems, 14(9), 3751-3778.

Kelemen, P. B., Kikawa, E., Miller, D. J., Abe, N., Bach, W., Carlson, R. L., ... & Takazawa, E. (2004). Proceedings of the Ocean Drilling Program, Initial Reports, Volume 209, Drilling mantle peridotite along the Mid-Atlantic Ridge from 14 degrees to 16 degrees N, Sites 1269-1275.

Richards, F. D., Hoggard, M. J., Cowton, L. R., & White, N. J. (2018). Reassessing the thermal structure of oceanic lithosphere with revised global inventories of basement depths and heat flow measurements. Journal of Geophysical Research: Solid Earth, 123(10), 9136-9161.

van Keken, P. E., Currie, C., King, S. D., Behn, M. D., Cagnioncle, A., He, J., ... & Wang, K. (2008). A community benchmark for subduction zone modeling. Physics of the Earth and Planetary Interiors, 171(1-4), 187-197.

Van Zelst, I., Crameri, F., Pusok, A. E., Glerum, A., Dannberg, J., & Thieulot, C. (2022). 101 geodynamic modelling: how to design, interpret, and communicate numerical studies of the solid Earth. Solid Earth, 13(3), 583-637.

---

## Referee Report (RR1)

Review on "The effect of temperature-dependent material properties on simple thermal models of subduction zones" by van Zelst et al.

The authors implement temperature-dependent thermal parameters in numerical models to test their effects on the thermal structure of subduction zones. I appreciate their efforts on pushing forward the community to use more sophisticated/realistic processes to simulate plate tectonics, and it's good to see the model changes affected by the temperature-dependent thermal parameters. However, I feel the strategy of this MS is a bit strange. On one hand, the authors claim that we should use more realistic/complex temperature-dependent thermal parameters in subduction simulation. However, on the other hand, the authors test the effect of the temperature-dependent thermal parameters with an extremely simplified model (not only the model setup, but also the governing equations). I do not believe that important physics could only be revealed by simplified models. I am fine with the current results presented in the MS, but would like to suggest the authors to use more sophisticated models for their future numerical studies.

My questions and comments are listed below.

- Temperature could affect brittle-ductile deformation, which in turn affects seismicity. Unfortunately, the authors do not provide any discussion. Maybe add some brief discussion in section 4.1.

- Lines 11-12: strange conclusion: "……temperature-dependent thermal parameters…… has a secondary effect on ……compared to…… ". I fully agree with the previous reviewer who also questioned this conclusion. This kind of expression is indeed very confusing. In my opinion, the aim of this study is to clearly show readers the effect of the temperature-dependent thermal parameters on subduction evolution. I guess the authors do not need to emphasize the competition in terms of importance between the thermal parameters and other model parameters (e.g., plate rheology or age).

- Line 105: the temperature equation. What are the "external heat sources"? Shear heating, for instance? If so, I really do not understand why, since you are dealing with temperature evolution. It's really unnecessary to exclude the heat sources (especially when we are able to implement).

- Equation 2: Please explain explicitly with several lines why do you use zero gravitational acceleration, even for the vertical direction (and do not just refer to the van Keken paper without giving any necessary explanations). Besides, does "zero gravitational acceleration" mean there is no pressure? Is subduction purely driven by the velocity boundary condition?

- Line 141: Do not like this expression: "as we do not aim to …'"

➢ Line 147 Could different overriding plate structures (thickness and thermal structure) affect the slab thermal structure?

➢ Fig. 1b: Do the authors use constant temperature for the asthenospheric mantle underneath the overriding plate? In my opinion, this is over-simplified, since it is not compex to implement the adiabatic temperature gradient (and it is more realistic).

➢ Line 180: Plate model also has the shortcoming to describe the thermal structure for young plate ages.

➢ Line 210, Figures S8-S17: Too many 2D figures. Please show the difference in a concise manner, e.g., plot difference with temperature profiles?

➢ Figure 2 "The lighter colors indicate the crustal approximation for the thermal conductivity (i.e., multiplied by 0.5) and the density (i.e., multiplied by 0.79)." and Line 295. Why do you use the approximation rather than the realistic thermal parameters for the crustal domain?

➢ Line 245 "……first-order effect……" is confusing.

➢ Fig. 6: It's good to plot the difference between the two models. However, except the thermal parameters, the T-profile left boundary is different in these two models, i.e., the ref. model using the half-space cooling model while the preferred model using the plate model. This is weird. One should keep all other parameters (except the three thermal parameters) identical. Alternatively, one could compare the case2c_bc model with the case2c_all model.

➢ Lines 387-389, 392-393, Figs. 6a-b: The main reason of the colder slab and the overring plate in the preferred model (case2c_all) is larger thermal diffusivity (e.g., Fig. 3)? The lower part of the slab in the preferred model (case2c_all) is slightly warmer, why? One should explain and section 3.5 is the right place.

➢ Figs. 6c-d: It is quite difficult to understand the initial setup of the case2c_all_cp model. Since this is quite an important model, why not describe it in Table 1? Furthermore, this model behaves much different from the case2c_all model, and one should provide description in section 3.5.

➢ Supplement: Do readers really need to read through the 29 figures? Are these figures really essential for readers to fully understand the main text?

---

## Author Response (AR2)

Dear Editor, dear Reviewers,

Please find enclosed our revised manuscript (Preprint egusphere-2022-768) on 'The effect of temperature-dependent material properties on simple thermal models of subduction zones' by Van Zelst et al.. We thank the two reviewers and the editor for their feedback. We addressed the - mostly textual - comments by the reviewers in our revised version (see below). Responses to the suggestions by the reviewers are indicated in green. Line numbers refer to the line numbers of the provided tracked changes file.

Thank you for considering this revised manuscript for publication.

Yours sincerely,
Iris van Zelst (corresponding author),
Cedric Thieulot, Timothy J. Craig

**Letter from the Editor**

Dear Authors,

Thank you for submitting the revised paper version, which has been assessed by two reviewers. As you will see, both reviewers are positive but raised some comments, which need to be addressed. Please answer the comments, revise the paper respectively and make sure that the final version you upload includes the missing data mentioned in the "Data availability" section: "The models were run with the open source code xFieldstone (which will be made available on GitHub upon paper acceptance; the exact version of the code will be stored in a Zenodo repository). Upon paper acceptance, we will also make a Zenodo repository with the data used to reproduce the van Keken et al. (2008) benchmark and the Matlab scripts used for the postprocessing of the results and generation of the figures"

With my best wishes.

Taras Gerya

We have addressed the comments from Reviewer 3 and made the code and the data and scripts available on Zenodo with appropriate updated references in the text.

**Referee #2**

Dear authors,

Thank you for addressing the comments raised. The revised manuscript is much improved and is now a factual account of the results presented.

Please make sure that the final accepted version you upload includes the missing data mentioned in the "Data availability" section:

"The models were run with the open source code xFieldstone (which will be made available on GitHub upon paper acceptance; the exact version of the code will be stored in a Zenodo repository). Upon paper acceptance, we will also make a Zenodo repository with the data used to reproduce the van Keken et al. (2008) benchmark and the Matlab scripts used for the postprocessing of the results and generation of the figures"

Thanks! We have now made the code, the data, and the scripts available on Zenodo and added the references in the text.

**Referee #3**

The authors implement temperature-dependent thermal parameters in numerical models to test their effects on the thermal structure of subduction zones. I appreciate their efforts on pushing forward the community to use more sophisticated/realistic processes to simulate plate tectonics, and it's good to see the model changes affected by the temperature-dependent thermal parameters. However, I feel the strategy of this MS is a bit strange. On one hand, the authors claim that we should use more realistic/complex temperature-dependent thermal parameters in subduction simulation. However, on the other hand, the authors test the effect of the temperature-dependent thermal parameters with an extremely simplified model (not only the model setup, but also the governing equations). I do not believe that important physics could only be revealed by simplified models. I am fine with the current results presented in the MS, but would like to suggest the authors to use more sophisticated models for their future numerical studies.

My questions and comments are listed below.

- Temperature could affect brittle-ductile deformation, which in turn affects seismicity. Unfortunately, the authors do not provide any discussion. Maybe add some brief discussion in section 4.1.

  We have expanded the discussion on megathrust seismicity and the brittle-ductile ductile transition by adding three paragraphs to Section 4.1.3 (lines 533 - 552).

- Lines 11-12: strange conclusion: "......temperature-dependent thermal parameters...... has a secondary effect on ......compared to...... ". I fully agree with the previous reviewer who also questioned this conclusion. This kind of expression is indeed very confusing. In my opinion, the aim of this study is to clearly show readers the effect of the temperature-dependent thermal parameters on subduction evolution. I guess the authors do not need to emphasize the competition in terms of importance between the thermal parameters and other model parameters (e.g., plate rheology or age).

We initially wrote this to comply with previous reviewers' comments who wanted to us to be clear that temperature-dependent parameters are not the main first-order parameters that could affect thermal structure. However, we agree that the way it is written here is stunted and unnecessary. The rest of the manuscript nuances the point of the other reviewers sufficiently. We have therefore rephrased this part of the abstract (lines 10 - 13).

- Line 105: the temperature equation. What are the "external heat sources"? Shear heating, for instance? If so, I really do not understand why, since you are dealing with temperature evolution. It's really unnecessary to exclude the heat sources (especially when we are able to implement).

We now specify what kind of external heat sources we exclude in lines 108 - 109 and why we exclude them: to simplify the model. Mainly, the reasoning is that we stay as close to the model setup of Van Keken et al. (2008) as possible, who also excluded external heat sources, such that we can easily compare our model results and the effect of temperature-dependent thermal parameters in a well-defined, simple model setup. We briefly discuss the potential effect of external heat sources in the model in the discussion.

- Equation 2: Please explain explicitly with several lines why do you use zero gravitational acceleration, even for the vertical direction (and do not just refer to the van Keken paper without giving any necessary explanations). Besides, does "zero gravitational acceleration" mean there is no pressure? Is subduction purely driven by the velocity boundary condition?

Indeed, the subduction is purely driven by the velocity boundary condition (i.e., kinematically-driven subduction). We added a few lines to explain why we use zero gravitational acceleration (lines 105 - 107):

"In this formulation of the Stokes equations, we implicitly assume zero gravitational acceleration such that we have purely  kinematically-driven subduction. This allows us to have the same (fixed) subduction geometry in all models."

- Line 141: Do not like this expression: "as we do not aim to ...'"

Since none of the other reviewers commented on this and the scientific content of the manuscript isn't affected by this phrasing, we have kept the expression as is.

- Line 147 Could different overriding plate structures (thickness and thermal structure) affect the slab thermal structure?

  We did not test this, but - to an extent - the overriding plate would probably indeed affect the slab thermal structure, especially when the thickness of the overriding plate changes and hence the mantle wedge corner is shifted to another location. We comment on this in the discussion and we added a sentence on the potential effect of the thickness of the overriding plate (line 632).

- Fig. 1b: Do the authors use constant temperature for the asthenospheric mantle underneath the overriding plate? In my opinion, this is over-simplified, since it is not complex to implement the adiabatic temperature gradient (and it is more realistic).

  We do not define the temperature in the mantle beneath the overriding plate. Instead, we prescribe that the temperature of incoming material is at 1300°C. Besides that, the temperature of the mantle wedge is then solved for according to the prescribed temperature boundary conditions (see Figure 1a).

- Line 180: Plate model also has the shortcoming to describe the thermal structure for young plate ages.

  This is correct. However, we do not consider the plate model in our paper and therefore do not mention it. Instead, as we are basing ourselves on Van Keken et al. (2008), we merely comment on the shortcomings of their use of the half-space cooling model. Using the half-space cooling model is fine for the benchmark they suggested, but is too simplistic when considering more realistic setups.

- Line 210, Figures S8-S17: Too many 2D figures. Please show the difference in a concise manner, e.g., plot difference with temperature profiles?

  The difference between the temperature profiles is already shown in Figure S18. We now also refer to this figure in the text for the reader's convenience.

- Figure 2 "The lighter colors indicate the crustal approximation for the thermal conductivity (i.e., multiplied by 0.5) and the density (i.e., multiplied by 0.79)." and Line 295. Why do you use the approximation rather than the realistic thermal parameters for the crustal domain?

  We added the set of models with a rough crustal approximation to determine the first-order effect a crust would have on the thermal structure of a subduction zone in our models. This was done after previous reviewers asked about the potential effect of a

crust in our models. Since implementing the full, complete set of realistic thermal parameters for the crustal domain is beyond the scope of this paper, we opted for this simple approximation instead to give the reader (and ourselves) a sense of how (and if) different crustal parameters might change the results.

We now added a line where we introduce the simplified approximation of a crustal layer to direct the reader to the discussion where we discuss the implications of this simplification and alternatives and considerations (line 289).

- Line 245 "......first-order effect......" is confusing.

We removed 'first-order'.

- Fig. 6: It's good to plot the difference between the two models. However, except the thermal parameters, the T-profile left boundary is different in these two models, i.e., the ref. model using the half-space cooling model while the preferred model using the plate model. This is weird. One should keep all other parameters (except the three thermal parameters) identical. Alternatively, one could compare the case2c_bc model with the case2c_all model.

We showed the difference between these two models, because we wanted to compare the models with the biggest difference, i.e., model case2c_PvK and case2c_all (which are also the models shown in the main paper in Figure 4 and 5). We agree this is perhaps not the most straightforward comparison, but as the differences between case2c_PvK and case2c_bc are very minor (see Figure 7, Table 2, and Section 3.1) it does not really matter whether we use case2c_PvK or case2c_bc for this comparison figure. Since the other reviewers did not comment on this, we keep this figure as is.

- Lines 387-389, 392-393, Figs. 6a-b: The main reason of the colder slab and the overring plate in the preferred model (case2c_all) is larger thermal diffusivity (e.g., Fig. 3)? The lower part of the slab in the preferred model (case2c_all) is slightly warmer, why? One should explain and section 3.5 is the right place.

We added the following explanation to this section (lines 406 - 409):

"This difference in slab temperature between the two models is partly due to the difference in boundary condition, as the plate cooling model in case2c_all is cooler than the half-space cooling model of case2c_PvK at shallow depths due to increased conductivity at low temperatures, and warmer at larger depth due to the imposition of a lower thermal boundary condition in the plate model. This also depends on the age of the subducting slab (see Section 3.6)."

- Figs. 6c-d: It is quite difficult to understand the initial setup of the case2c_all_cp model. Since this is quite an important model, why not describe it in Table 1? Furthermore, this

model behaves much different from the case2c_all model, and one should provide description in section 3.5.

Model case2c_all_cp is described in Table 1 in the note below the table and explained in the text in Section 2.6 (lines 301 - 303).

In section 3.5 we only discuss the model without the continental plate parameterisation, which is why we do not comment on model case2c_all_cp. The confusion likely resulted from the fact that we refer to the entirety of Figure 6, instead of just the first two panels a and b. We have now changed this and in Section 3.8 (where we discuss the results of the models with the parameterised overriding plate) we now refer to Figure 6 c and d.

- Supplement: Do readers really need to read through the 29 figures? Are these figures really essential for readers to fully understand the main text?

We include all the figures in the supplementary material such that readers have access to all the results. Previous reviewers did not mind the amount of figures in the supplementary material and in fact many of the figures in the supplementary material were added because of suggestions by the reviewers, so we keep the supplementary material as is.